# CALIBRATION TESTS BEYOND CLASSIFICATION

**David Widmann**
Department of Information Technology
Uppsala University, Sweden
`david.widmann@it.uu.se`

**Fredrik Lindsten**
Division of Statistics and Machine Learning
Linköping University, Sweden
`fredrik.lindsten@liu.se`

**Dave Zachariah**
Department of Information Technology
Uppsala University, Sweden
`dave.zachariah@it.uu.se`

## ABSTRACT

Most supervised machine learning tasks are subject to irreducible prediction errors. Probabilistic predictive models address this limitation by providing probability distributions that represent a belief over plausible targets, rather than point estimates. Such models can be a valuable tool in decision-making under uncertainty, provided that the model output is meaningful and interpretable. Calibrated models guarantee that the probabilistic predictions are neither over- nor under-confident. In the machine learning literature, different measures and statistical tests have been proposed and studied for evaluating the calibration of classification models. For regression problems, however, research has been focused on a weaker condition of calibration based on predicted quantiles for real-valued targets. In this paper, we propose the first framework that unifies calibration evaluation and tests for general probabilistic predictive models. It applies to any such model, including classification and regression models of arbitrary dimension. Furthermore, the framework generalizes existing measures and provides a more intuitive reformulation of a recently proposed framework for calibration in multi-class classification. In particular, we reformulate and generalize the kernel calibration error, its estimators, and hypothesis tests using scalar-valued kernels, and evaluate the calibration of real-valued regression problems.[1]

## 1 INTRODUCTION

We consider the general problem of modelling the relationship between a feature $X$ and a target $Y$ in a probabilistic setting, i.e., we focus on models that approximate the conditional probability distribution $\mathbb{P}(Y|X)$ of target $Y$ for given feature $X$. The use of probabilistic models that output a probability distribution instead of a point estimate demands guarantees on the predictions beyond accuracy, enabling meaningful and interpretable predicted uncertainties. One such statistical guarantee is calibration, which has been studied extensively in metereological and statistical literature (DeGroot & Fienberg, 1983; Murphy & Winkler, 1977).

A calibrated model ensures that almost every prediction matches the conditional distribution of targets given this prediction. Loosely speaking, in a classification setting a predicted distribution of the model is called calibrated (or reliable), if the empirically observed frequencies of the different classes match the predictions in the long run, if the same class probabilities would be predicted repeatedly. A classical example is a weather forecaster who predicts each day if it is going to rain on the next day. If she predicts rain with probability 60% for a long series of days, her forecasting model is calibrated *for predictions of 60%* if it actually rains on 60% of these days.

If this property holds for almost every probability distribution that the model outputs, then the model is considered to be calibrated. Calibration is an appealing property of a probabilistic model since it

---

[1]The source code of the experiments is available at `https://github.com/devmotion/Calibration_ICLR2021`.

provides safety guarantees on the predicted distributions even in the common case when the model does not predict the true distributions $\mathbb{P}(Y|X)$. Calibration, however, does not guarantee accuracy (or refinement)—a model that always predicts the marginal probabilities of each class is calibrated but probably inaccurate and of limited use. On the other hand, accuracy does not imply calibration either since the predictions of an accurate model can be too over-confident and hence miscalibrated, as observed, e.g., for deep neural networks (Guo et al., 2017).

In the field of machine learning, calibration has been studied mainly for classification problems (Bröcker, 2009; Guo et al., 2017; Kull et al., 2017; 2019; Kumar et al., 2018; Platt, 2000; Vaicenavicius et al., 2019; Widmann et al., 2019; Zadrozny, 2002) and for quantiles and confidence intervals of models for regression problems with real-valued targets (Fasiolo et al., 2020; Ho & Lee, 2005; Kuleshov et al., 2018; Rueda et al., 2006; Taillardat et al., 2016). In our work, however, we do not restrict ourselves to these problem settings but instead consider calibration for arbitrary predictive models. Thus, we generalize the common notion of calibration as:

**Definition 1.** Consider a model $P_X := P(Y|X)$ of a conditional probability distribution $\mathbb{P}(Y|X)$. Then model $P$ is said to be calibrated if and only if

$$\mathbb{P}(Y|P_X) = P_X \qquad \text{almost surely.} \qquad (1)$$

If $P$ is a classification model, Definition 1 coincides with the notion of (multi-class) calibration by Bröcker (2009); Kull et al. (2019); Vaicenavicius et al. (2019). Alternatively, in classification some authors (Guo et al., 2017; Kumar et al., 2018; Naeini et al., 2015) study the strictly weaker property of confidence calibration (Kull et al., 2019), which only requires

$$\mathbb{P}\left(Y = \arg\max P_X \,\middle|\, \max P_X\right) = \max P_X \qquad \text{almost surely.} \qquad (2)$$

This notion of calibration corresponds to calibration according to Definition 1 for a reduced problem with binary targets $\widetilde{Y} := \mathbb{1}(Y = \arg\max P_X)$ and Bernoulli distributions $\widetilde{P}_X := \mathrm{Ber}(\max P_X)$ as probabilistic models.

For real-valued targets, Definition 1 coincides with the so-called distribution-level calibration by Song et al. (2019). Distribution-level calibration implies that the predicted quantiles are calibrated, i.e., the outcomes for all real-valued predictions of the, e.g., 75% quantile are actually below the predicted quantile with 75% probability (Song et al., 2019, Theorem 1). Conversely, although quantile-based calibration is a common approach for real-valued regression problems (Fasiolo et al., 2020; Ho & Lee, 2005; Kuleshov et al., 2018; Rueda et al., 2006; Taillardat et al., 2016), it provides weaker guarantees on the predictions. For instance, the linear regression model in Fig. 1 empirically shows quantiles that appear close to being calibrated albeit being uncalibrated according to Definition 1.

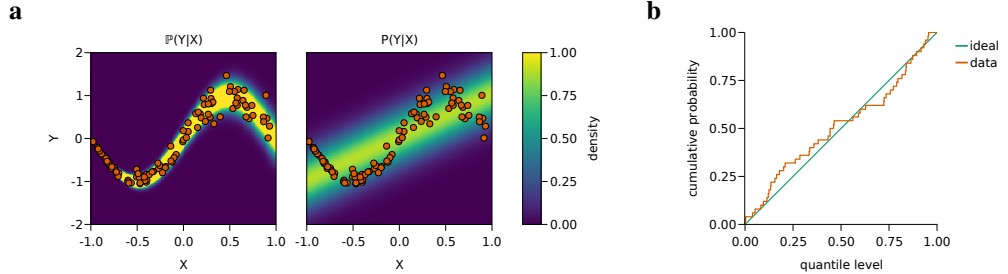

Figure 1: Illustration of a conditional distribution $\mathbb{P}(Y|X)$ with scalar feature and target. We consider a Gaussian predictive model $P$, obtained by ordinary least squares regression with 100 training data points (orange dots). Empirically the predicted quantiles on 50 validation data points appear close to being calibrated, although model $P$ is uncalibrated according to Definition 1. Using the framework in this paper, on the same validation data a statistical test allows us to reject the null hypothesis that model $P$ is calibrated at a significance level of $\alpha = 0.05$ ($p < 0.05$). See Appendix A.1 for details.

Figure 1 also raises the question of how to assess calibration for general target spaces in the sense of Definition 1, without having to rely on visual inspection. In classification, measures of calibration such as the commonly used expected calibration error (ECE) (Guo et al., 2017; Kull et al., 2019;

Naeini et al., 2015; Vaicenavicius et al., 2019) and the maximum calibration error (MCE) (Naeini et al., 2015) try to capture the average and maximal discrepancy between the distributions on the left hand side and the right hand side of Eq. (1) or Eq. (2), respectively. These measures can be generalized to other target spaces (see Definition B.1), but unfortunately estimating these calibration errors from observations of features and corresponding targets is problematic. Typically, the predictions are different for (almost) all observations, and hence estimation of the conditional probability $\mathbb{P}(Y|P_X)$, which is needed in the estimation of ECE and MCE, is challenging even for low-dimensional target spaces and usually leads to biased and inconsistent estimators (Vaicenavicius et al., 2019).

Kernel-based calibration errors such as the maximum mean calibration error (MMCE) (Kumar et al., 2018) and the kernel calibration error (KCE) (Widmann et al., 2019) for confidence and multi-class calibration, respectively, can be estimated without first estimating the conditional probability and hence avoid this issue. They are defined as the expected value of a weighted sum of the differences of the left and right hand side of Eq. (1) for each class, where the weights are given as a function of the predictions (of all classes) and chosen such that the calibration error is maximized. A reformulation with matrix-valued kernels (Widmann et al., 2019) yields unbiased and differentiable estimators without explicit dependence on $\mathbb{P}(Y|P_X)$, which simplifies the estimation and allows to explicitly account for calibration in the training objective (Kumar et al., 2018). Additionally, the kernel-based framework allows the derivation of reliable statistical hypothesis tests for calibration in multi-class classification (Widmann et al., 2019).

However, both the construction as a weighted difference of the class-wise distributions in Eq. (1) and the reformulation with matrix-valued kernels require finite target spaces and hence cannot be applied to regression problems. To be able to deal with general target spaces, we present a new and more general framework of calibration errors without these limitations.

Our framework can be used to reason about and test for calibration of *any probabilistic predictive model*. As explained above, this is in stark contrast with existing methods that are restricted to simple output distributions, such as classification and *scalar-valued* regression problems. A *key contribution* of this paper is a new framework that is applicable to *multivariate* regression, as well as situations when the output is of a different (e.g., discrete ordinal) or more complex (e.g., graph-structured) type, with clear practical implications.

Within this framework a KCE for general target spaces is obtained. We want to highlight that for multi-class classification problems its formulation is more intuitive and simpler to use than the measure proposed by Widmann et al. (2019) based on matrix-valued kernels. To ease the application of the KCE we derive several estimators of the KCE with subquadratic sample complexity and their asymptotic properties in tests for calibrated models, which improve on existing estimators and tests in the two-sample test literature by exploiting the special structure of the calibration framework. Using the proposed framework, we numerically evaluate the calibration of neural network models and ensembles of such models.

## 2 CALIBRATION ERROR: A GENERAL FRAMEWORK

In classification, the distributions on the left and right hand side of Eq. (1) can be interpreted as vectors in the probability simplex. Hence ultimately the distance measure for ECE and MCE (see Definition B.1) can be chosen as a distance measure of real-valued vectors. The total variation, Euclidean, and squared Euclidean distances are common choices (Guo et al., 2017; Kull et al., 2019; Vaicenavicius et al., 2019). However, in a general setting measuring the discrepancy between $\mathbb{P}(Y|P_X)$ and $P_X$ cannot necessarily be reduced to measuring distances between vectors. The conditional distribution $\mathbb{P}(Y|P_X)$ can be arbitrarily complex, even if the predicted distributions are restricted to a simple class of distributions that can be represented as real-valued vectors. Hence in general we have to resort to dedicated distance measures of probability distributions.

Additionally, the estimation of conditional distributions $\mathbb{P}(Y|P_X)$ is challenging, even more so than in the restricted case of classification, since in general these distributions can be arbitrarily complex. To circumvent this problem, we propose to use the following construction: We define a random variable $Z_X \sim P_X$ obtained from the predictive model and study the discrepancy between the *joint* distributions of the two pairs of random variables $(P_X, Y)$ and $(P_X, Z_X)$, respectively, instead of

the discrepancy between the *conditional* distributions $\mathbb{P}(Y|P_X)$ and $P_X$. Since

$$(P_X, Y) \stackrel{d}{=} (P_X, Z_X) \qquad \text{if and only if} \qquad \mathbb{P}(Y|P_X) = P_X \quad \text{almost surely,}$$

model $P$ is calibrated if and only if the distributions of $(P_X, Y)$ and $(P_X, Z_X)$ are equal.

The random variable pairs $(P_X, Y)$ and $(P_X, Z_X)$ take values in the product space $\mathcal{P} \times \mathcal{Y}$, where $\mathcal{P}$ is the space of predicted distributions $P_X$ and $\mathcal{Y}$ is the space of targets $Y$. For instance, in classification, $\mathcal{P}$ could be the probability simplex and $\mathcal{Y}$ the set of all class labels, whereas in the case of Gaussian predictive models for scalar targets $\mathcal{P}$ could be the space of normal distributions and $\mathcal{Y}$ be $\mathbb{R}$.

The study of the joint distributions of $(P_X, Y)$ and $(P_X, Z_X)$ motivates the definition of a generally applicable calibration error as an integral probability metric (Müller, 1997; Sriperumbudur et al., 2009; 2012) between these distributions. In contrast to common $f$-divergences such as the Kullback-Leibler divergence, integral probability metrics do not require that one distribution is absolutely continuous with respect to the other, which cannot be guaranteed in general.

---

**Definition 2.** Let $\mathcal{Y}$ denote the space of targets $Y$, and $\mathcal{P}$ the space of predicted distributions $P_X$. We define the calibration error with respect to a space of functions $\mathcal{F}$ of the form $f\colon \mathcal{P} \times \mathcal{Y} \to \mathbb{R}$ as

$$\mathrm{CE}_{\mathcal{F}} := \sup_{f \in \mathcal{F}} \big| \mathbb{E}_{P_X, Y} f(P_X, Y) - \mathbb{E}_{P_X, Z_X} f(P_X, Z_X) \big|. \tag{3}$$

---

By construction, if model $P$ is calibrated, then $\mathrm{CE}_{\mathcal{F}} = 0$ regardless of the choice of $\mathcal{F}$. However, the converse statement is not true for arbitrary function spaces $\mathcal{F}$. From the theory of integral probability metrics (see, e.g., Müller, 1997; Sriperumbudur et al., 2009; 2012), we know that for certain choices of $\mathcal{F}$ the calibration error in Eq. (3) is a well-known metric on the product space $\mathcal{P} \times \mathcal{Y}$, which implies that $\mathrm{CE}_{\mathcal{F}} = 0$ if and only if model $P$ is calibrated. Prominent examples include the maximum mean discrepancy[2] (MMD) (Gretton et al., 2007), the total variation distance, the Kantorovich distance, and the Dudley metric (Dudley, 1989, p. 310).

As pointed out above, Definition 2 is a generalization of the definition for multi-class classification proposed by Widmann et al. (2019)—which is based on vector-valued functions and only applicable to finite target spaces—to *any probabilistic predictive model*. In Appendix E we show this explicitly and discuss the special case of classification problems in more detail. Previous results (Widmann et al., 2019) imply that in classification MMCE and, for common distance measures $d(\cdot, \cdot)$ such as the total variation and squared Euclidean distance, $\mathrm{ECE}_d$ and $\mathrm{MCE}_d$ are special cases of $\mathrm{CE}_{\mathcal{F}}$. In Appendix G we show that our framework also covers natural extensions of $\mathrm{ECE}_d$ and $\mathrm{MCE}_d$ to countably infinite discrete target spaces, which to our knowledge have not been studied before and occur, e.g., in Poisson regression.

The literature of integral probability metrics suggests that we can resort to estimating $\mathrm{CE}_{\mathcal{F}}$ from i.i.d. samples from the distributions of $(P_X, Y)$ and $(P_X, Z_X)$. For the MMD, the Kantorovich distance, and the Dudley metric tractable strongly consistent empirical estimators exist (Sriperumbudur et al., 2012). Here the empirical estimator for the MMD is particularly appealing since compared with the other estimators "it is computationally cheaper, the empirical estimate converges at a faster rate to the population value, and the rate of convergence is independent of the dimension $d$ of the space (for $S = \mathbb{R}^d$)" (Sriperumbudur et al. (2012)).

Our specific design of $(P_X, Z_X)$ can be exploited to improve on these estimators. If $\mathbb{E}_{Z_x \sim P_x} f(P_x, Z_x)$ can be evaluated analytically for a fixed prediction $P_x$, then $\mathrm{CE}_{\mathcal{F}}$ can be estimated empirically with reduced variance by marginalizing out $Z_X$. Otherwise $\mathbb{E}_{Z_x \sim P_x} f(P_x, Z_x)$ has to be estimated, but in contrast to the common estimators of the integral probability metrics discussed above the artificial construction of $Z_X$ allows us to approximate it by numerical integration methods such as (quasi) Monte Carlo integration or quadrature rules with arbitrarily small error and variance. Monte Carlo integration preserves statistical properties of the estimators such as unbiasedness and consistency.

---

[2]As we discuss in Section 3, the MMD is a metric if and only if the employed kernel is characteristic.

# 3 KERNEL CALIBRATION ERROR

For the remaining parts of the paper we focus on the MMD formulation of $\mathrm{CE}_{\mathcal{F}}$ due to the appealing properties of the common empirical estimator mentioned above. We derive calibration-specific analogues of results for the MMD that exploit the special structure of the distribution of $(P_X, Z_X)$ to improve on existing estimators and tests in the MMD literature. To the best of our knowledge these variance-reduced estimators and tests have not been discussed in the MMD literature.

Let $k \colon (\mathcal{P} \times \mathcal{Y}) \times (\mathcal{P} \times \mathcal{Y}) \to \mathbb{R}$ be a measurable kernel with corresponding reproducing kernel Hilbert space (RKHS) $\mathcal{H}$, and assume that

$$\mathbb{E}_{P_X,Y}\, k^{1/2}\big((P_X, Y), (P_X, Y)\big) < \infty \quad \text{and} \quad \mathbb{E}_{P_X,Z_X}\, k^{1/2}\big((P_X, Z_X), (P_X, Z_X)\big) < \infty.$$

We discuss how such kernels can be constructed in a generic way in Section 3.1 below.

**Definition 3.** Let $\mathcal{F}_k$ denote the unit ball in $\mathcal{H}$, i.e., $\mathcal{F} \coloneqq \{f \in \mathcal{H} | \|f\|_{\mathcal{H}} \leq 1\}$. Then the kernel calibration error (KCE) with respect to kernel $k$ is defined as

$$\mathrm{KCE}_k \coloneqq \mathrm{CE}_{\mathcal{F}_k} = \sup_{f \in \mathcal{F}_k} \big|\mathbb{E}_{P_X,Y} f(P_X, Y) - \mathbb{E}_{P_X,Z_X} f(P_X, Z_X)\big|.$$

As known from the MMD literature, a more explicit formulation can be given for the squared kernel calibration error $\mathrm{SKCE}_k \coloneqq \mathrm{KCE}_k^2$ (see Lemma B.2). A similar explicit expression for $\mathrm{SKCE}_k$ was obtained by Widmann et al. (2019) for the special case of classification problems. However, their expression relies on $\mathcal{Y}$ being finite and is based on matrix-valued kernels over the finite-dimensional probability simplex $\mathcal{P}$. A key difference to the expression in Lemma B.2 is that we instead propose to use real-valued kernels defined on the product space of predictions and targets. This construction is applicable to arbitrary target spaces and does not require $\mathcal{Y}$ to be finite.

## 3.1 CHOICE OF KERNEL

The construction of the product space $\mathcal{P} \times \mathcal{Y}$ suggests the use of tensor product kernels $k = k_{\mathcal{P}} \otimes k_{\mathcal{Y}}$, where $k_{\mathcal{P}} \colon \mathcal{P} \times \mathcal{P} \to \mathbb{R}$ and $k_{\mathcal{Y}} \colon \mathcal{Y} \times \mathcal{Y} \to \mathbb{R}$ are kernels on the spaces of predicted distributions and targets, respectively.[3]

By definition, so-called characteristic kernels guarantee that $\mathrm{KCE} = 0$ if and only if the distributions of $(P_X, Y)$ and $(P_X, Z_X)$ are equal (Fukumizu et al., 2004; 2008). Many common kernels such as the Gaussian and Laplacian kernel on $\mathbb{R}^d$ are characteristic (Fukumizu et al., 2008).[4] Szabó & Sriperumbudur (2018, Theorem 4) showed that a tensor product kernel $k_{\mathcal{P}} \otimes k_{\mathcal{Y}}$ is characteristic if $k_{\mathcal{P}}$ and $k_{\mathcal{Y}}$ are characteristic, continuous, bounded, and translation-invariant kernels on $\mathbb{R}^d$, but the implication does not hold for general characteristic kernels (Szabó & Sriperumbudur, 2018, Example 1). For calibration evaluation, however, it is sufficient to be able to distinguish between the conditional distributions $\mathbb{P}(Y|P_X)$ and $\mathbb{P}(Z_X|P_X) = P_X$. Therefore, in contrast to the regular MMD setting, it is *sufficient that kernel $k_{\mathcal{Y}}$ is characteristic and kernel $k_{\mathcal{P}}$ is non-zero almost surely*, to guarantee that $\mathrm{KCE} = 0$ if and only if model $P$ is calibrated. Thus it is suggestive to construct kernels on general spaces of predicted distributions as

$$k_{\mathcal{P}}(p, p') = \exp\big(-\lambda d_{\mathcal{P}}^{\nu}(p, p')\big), \tag{4}$$

where $d_{\mathcal{P}}(\cdot, \cdot)$ is a metric on $\mathcal{P}$ and $\nu, \lambda > 0$ are kernel hyperparameters. The Wasserstein distance is a widely used metric for distributions from optimal transport theory that allows to lift a ground metric on the target space and possesses many important properties (see, e.g., Peyré & Cuturi, 2019, Chapter 2.4). In general, however, it does not lead to valid kernels $k_{\mathcal{P}}$, apart from the notable exception of elliptically contoured distributions such as normal and Laplace distributions (Peyré & Cuturi, 2019, Chapter 8.3).

---

[3]As mentioned above, our framework rephrases and generalizes the construction used by Widmann et al. (2019). The matrix-valued kernels that they employ can be recovered by setting $k_{\mathcal{P}}$ to a Laplacian kernel on the probability simplex and $k_{\mathcal{Y}}(y, y') = \delta_{y,y'}$.

[4]For a general discussion about characteristic kernels and their relation to universal kernels we refer to the paper by Sriperumbudur et al. (2011).

In machine learning, common probabilistic predictive models output parameters of distributions such as mean and variance of normal distributions. Naturally these parameterizations give rise to injective mappings $\phi \colon \mathcal{P} \to \mathbb{R}^d$ that can be used to define a Hilbertian metric

$$d_{\mathcal{P}}(p, p') = \|\phi(p) - \phi(p')\|_2.$$

For such metrics, $k_{\mathcal{P}}$ in Eq. (4) is a valid kernel for all $\lambda > 0$ and $\nu \in (0, 2]$ (Berg et al., 1984, Corollary 3.3.3, Proposition 3.2.7). In Appendix D.3 we show that for many mixture models, and hence model ensembles, Hilbertian metrics between model components can be lifted to Hilbertian metrics between mixture models. This construction is a generalization of the Wasserstein-like distance for Gaussian mixture models proposed by Chen et al. (2019; 2020); Delon & Desolneux (2020).

## 3.2 ESTIMATION

Let $(X_1, Y_1), \ldots, (X_n, Y_n)$ be a data set of features and targets which are i.i.d. according to the law of $(X, Y)$. Moreover, for notational brevity, for $(p, y), (p', y') \in \mathcal{P} \times \mathcal{Y}$ we let

$$\begin{aligned} h\big((p, y), (p', y')\big) := k\big((p, y), (p', y')\big) &- \mathbb{E}_{Z \sim p}\, k\big((p, Z), (p', y')\big) \\ &- \mathbb{E}_{Z' \sim p'}\, k\big((p, y), (p', Z')\big) + \mathbb{E}_{Z \sim p, Z' \sim p'}\, k\big((p, Z), (p', Z')\big). \end{aligned}$$

Note that in contrast to the regular MMD we marginalize out $Z$ and $Z'$. Similar to the MMD, there exist consistent estimators of the SKCE, both biased and unbiased.

**Lemma 1.** *The plug-in estimator of* $\mathrm{SKCE}_k$ *is non-negatively biased. It is given by*

$$\widehat{\mathrm{SKCE}}_k = \frac{1}{n^2} \sum_{i,j=1}^{n} h\big((P_{X_i}, Y_i), (P_{X_j}, Y_j)\big).$$

Inspired by the block tests for the regular MMD (Zaremba et al., 2013), we define the following class of unbiased estimators. Note that in contrast to $\widehat{\mathrm{SKCE}}_k$ they do not include terms of the form $h\big((P_{X_i}, Y_i), (P_{X_i}, Y_i)\big)$.

**Lemma 2.** *The block estimator of* $\mathrm{SKCE}_k$ *with block size* $B \in \{2, \ldots, n\}$, *given by*

$$\widehat{\mathrm{SKCE}}_{k,B} := \left\lfloor \frac{n}{B} \right\rfloor^{-1} \sum_{b=1}^{\lfloor n/B \rfloor} \binom{B}{2}^{-1} \sum_{(b-1)B < i < j \leq bB} h\big((P_{X_i}, Y_i), (P_{X_j}, Y_j)\big),$$

*is an unbiased estimator of* $\mathrm{SKCE}_k$.

The extremal estimator with $B = n$ is a so-called U-statistic of $\mathrm{SKCE}_k$ (Hoeffding, 1948; van der Vaart, 1998), and hence it is the minimum variance unbiased estimator. All presented estimators are consistent, i.e., they converge to $\mathrm{SKCE}_k$ almost surely as the number $n$ of data points goes to infinity. The sample complexity of $\widehat{\mathrm{SKCE}}_k$ and $\widehat{\mathrm{SKCE}}_{k,B}$ is $O(n^2)$ and $O(Bn)$, respectively.

## 3.3 CALIBRATION TESTS

A fundamental issue with calibration errors in general, including ECE, is that their empirical estimates do not provide an answer to the question if a model is actually calibrated. Even if the measure is guaranteed to be zero if and only if the model is calibrated, usually the estimates of calibrated models are non-zero due to randomness in the data and (possibly) the estimation procedure. In classification, statistical hypothesis tests of the null hypothesis

$$H_0 \colon \text{model } P \text{ is calibrated,}$$

so-called calibration tests, have been proposed as a tool for checking rigorously if $P$ is calibrated (Bröcker & Smith, 2007; Vaicenavicius et al., 2019; Widmann et al., 2019). For multi-class classification, Widmann et al. (2019) suggested calibration tests based on the asymptotic distributions of estimators of the previously formulated KCE. Although for finite data sets the asymptotic distributions are only approximations of the actual distributions of these estimators, in their experiments with 10 classes the resulting $p$-value approximations seemed reliable whereas $p$-values obtained by

so-called consistency resampling (Bröcker & Smith, 2007; Vaicenavicius et al., 2019) underestimated the $p$-value and hence rejected the null hypothesis too often (Widmann et al., 2019).

For fixed block sizes $\sqrt{\lfloor n/B \rfloor}\big(\widehat{\text{SKCE}}_{k,B} - \text{SKCE}_k\big) \xrightarrow{d} \mathcal{N}\big(0, \sigma_B^2\big)$ as $n \to \infty$, and, under $H_0$, $n\widehat{\text{SKCE}}_{k,n} \xrightarrow{d} \sum_{i=1}^{\infty} \lambda_i(Z_i - 1)$ as $n \to \infty$, where $Z_i$ are independent $\chi_1^2$ distributed random variables. See Appendix B for details and definitions of the involved constants. From these results one can derive calibration tests that extend and generalize the existing tests for classification problems, as explained in Remarks B.1 and B.2. Our formulation illustrates also the close connection of these tests to different two-sample tests (Gretton et al., 2007; Zaremba et al., 2013).

## 4 ALTERNATIVE APPROACHES

For two-sample tests, Chwialkowski et al. (2015) suggested the use of the so-called unnormalized mean embedding (UME) to overcome the quadratic sample complexity of the minimum variance unbiased estimator and its intractable asymptotic distribution. As we show in Appendix C, there exists an analogous measure of calibration, termed unnormalized calibration mean embedding (UCME), with a corresponding calibration mean embedding (CME) test.

As an alternative to our construction based on the joint distributions of $(P_X, Y)$ and $(P_X, Z_X)$, one could try to directly compare the conditional distributions $\mathbb{P}(Y|P_X)$ and $\mathbb{P}(Z_X|P_X) = P_X$. For instance, Ren et al. (2016) proposed the conditional MMD based on the so-called conditional kernel mean embedding (Song et al., 2009; 2013). However, as noted by Park & Muandet (2020), its common definition as operator between two RKHS is based on very restrictive assumptions, which are violated in many situations (see, e.g., Fukumizu et al., 2013, Footnote 4) and typically require regularized estimates. Hence, even theoretically, often the conditional MMD is "not an exact measure of discrepancy between conditional distributions" (Park & Muandet (2020)). In contrast, the maximum conditional mean discrepancy (MCMD) proposed in a concurrent work by Park & Muandet (2020) is a random variable derived from much weaker measure-theoretical assumptions. The MCMD provides a local discrepancy conditional on random predictions whereas KCE is a global real-valued summary of these local discrepancies.[5]

## 5 EXPERIMENTS

In our experiments we evaluate the computational efficiency and empirical properties of the proposed calibration error estimators and calibration tests on both calibrated and uncalibrated models. By means of a classic regression problem from statistics literature, we demonstrate that the estimators and tests can be used for the evaluation of calibration of neural network models and ensembles of such models. This section contains only an high-level overview of these experiments to conserve space but all experimental details are provided in Appendix A.

### 5.1 EMPIRICAL PROPERTIES AND COMPUTATIONAL EFFICIENCY

We evaluate error, variance, and computation time of calibration error estimators for calibrated and uncalibrated Gaussian predictive models in synthetic regression problems. The results empirically confirm the consistency of the estimators and the computational efficiency of the estimator with block size $B = 2$ which, however, comes at the cost of increased error and variance.

Additionally, we evaluate empirical test errors of calibration tests at a fixed significance level $\alpha = 0.05$. The evaluations, visualized in Fig. 2 for models with ten-dimensional targets, demonstrate empirically that the percentage of incorrect rejections of $H_0$ converges to the set significance level as the number of samples increases. Moreover, the results highlight the computational burden of the calibration test that estimates quantiles of the intractable asymptotic distribution of $n\widehat{\text{SKCE}}_{k,n}$ by bootstrapping.

---

[5]In our calibration setting, the MCMD is almost surely equal to $\sup_{f \in \mathcal{F}_{\mathcal{Y}}} \big| \mathbb{E}_{Y|P_X}\big(f(Y)|P_X\big) - \mathbb{E}_{Z_X|P_X}\big(f(Z_X)|P_X\big)\big|$, where $\mathcal{F}_{\mathcal{Y}} := \{f: \mathcal{Y} \to \mathbb{R} \| \|f\|_{\mathcal{H}_{\mathcal{Y}}} \leq 1\}$ for an RKHS $\mathcal{H}_{\mathcal{Y}}$ with kernel $k_{\mathcal{Y}}: \mathcal{Y} \times \mathcal{Y} \to \mathbb{R}$. If kernel $k_{\mathcal{Y}}$ is characteristic, $\text{MCMD} = 0$ almost surely if and only if model $P$ is calibrated (Park & Muandet, 2020, Theorem 3.7). Although the definition of MCMD only requires a kernel $k_{\mathcal{Y}}$ on the target space, a kernel $k_{\mathcal{P}}$ on the space of predictions has to be specified for the evaluation of its regularized estimates.

As expected, due to the larger variance of $\widehat{\mathrm{SKCE}}_{k,2}$ the test with fixed block size $B = 2$ shows a decreased test power although being computationally much more efficient.

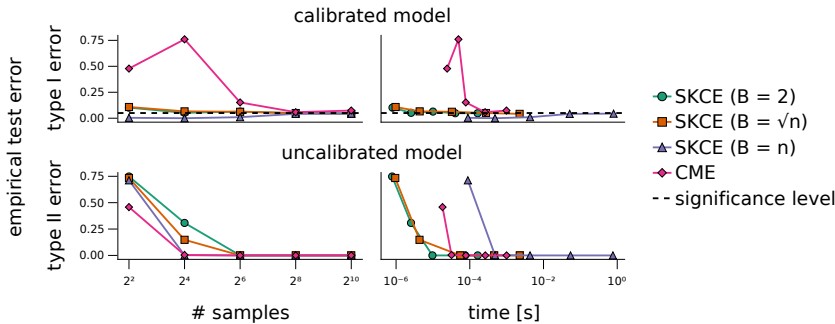

Figure 2: Empirical test errors for 500 data sets of $n \in \{4, 16, 64, 256, 1024\}$ samples from models with targets of dimension $d = 10$. The dashed black line indicates the set signficance level $\alpha = 0.05$.

## 5.2 Friedman 1 regression problem

The Friedman 1 regression problem (Friedman, 1979; 1991; Friedman et al., 1983) is a classic non-linear regression problem with ten-dimensional features and real-valued targets with Gaussian noise. We train a Gaussian predictive model whose mean is modelled by a shallow neural network and a single scalar variance parameter (consistent with the data-generating model) ten times with different initial parameters. Figure 3 shows estimates of the mean squared error (MSE), the average negative log-likelihood (NLL), $\mathrm{SKCE}_k$, and a $p$-value approximation for these models and their ensemble on the training and a separate test data set. All estimates indicate consistently that the models are overfit after 1500 training iterations. The estimations of $\mathrm{SKCE}_k$ and the $p$-values allow to focus on calibration specifically, whereas MSE indicates accuracy only and NLL, as any proper scoring rule (Bröcker, 2009), provides a summary of calibration and accuracy. The estimation of $\mathrm{SKCE}_k$ in addition to NLL could serve as another source of information for early stopping and model selection.

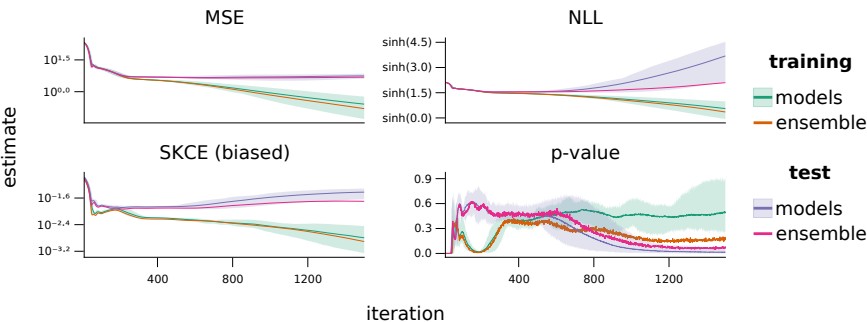

Figure 3: Mean squared error (MSE), average negative log-likelihood (NLL), $\widehat{\mathrm{SKCE}}_k$ (SKCE (biased)), and $p$-value approximation ($p$-value) of ten Gaussian predictive models for the Friedman 1 regression problem versus the number of training iterations. Evaluations on the training data set (100 samples) are displayed in green and orange, and on the test data set (50 samples) in blue and purple. The green and blue line and their surrounding bands represent the mean and the range of the evaluations of the ten models. The orange and purple lines visualize the evaluations of their ensemble.

## 6 CONCLUSION

We presented a framework of calibration estimators and tests for any probabilistic model that captures both classification and regression problems of arbitrary dimension as well as other predictive models. We successfully applied it for measuring calibration of (ensembles of) neural network models.

Our framework highlights connections of calibration to two-sample tests and optimal transport theory which we expect to be fruitful for future research. For instance, the power of calibration tests could be improved by heuristics and theoretical results about suitable kernel choices or hyperparameters (cf. Jitkrittum et al., 2016). It would also be interesting to investigate alternatives to KCE captured by our framework, e.g., by exploiting recent advances in optimal transport theory (cf. Genevay et al., 2016).

Since the presented estimators of $\mathrm{SKCE}_k$ are differentiable, we imagine that our framework could be helpful for improving calibration of predictive models, during training (cf. Kumar et al., 2018) or post-hoc. Currently, many calibration methods (see, e.g., Guo et al., 2017; Kull et al., 2019; Song et al., 2019) are based on optimizing the log-likelihood since it is a strictly proper scoring rule and thus encourages *both* accurate and reliable predictions. However, as for any proper scoring rule, "Per se, it is impossible to say how the score will rank unreliable forecast schemes [...]. The lack of reliability of one forecast scheme might be outbalanced by the lack of resolution of the other" (Bröcker (2009)). In other words, if one does not use a calibration method such as temperature scaling (Guo et al., 2017) that keeps accuracy invariant[6], it is unclear if the resulting model is trading off calibration for accuracy when using log-likelihood for re-calibration. Thus hypothetically flexible calibration methods might benefit from using the presented calibration error estimators.

### ACKNOWLEDGMENTS

We thank the reviewers for all the constructive feedback on our paper. This research is financially supported by the Swedish Research Council via the projects *Learning of Large-Scale Probabilistic Dynamical Models* (contract number: 2016-04278), *Counterfactual Prediction Methods for Heterogeneous Populations* (contract number: 2018-05040), and *Handling Uncertainty in Machine Learning Systems* (contract number: 2020-04122), by the Swedish Foundation for Strategic Research via the project *Probabilistic Modeling and Inference for Machine Learning* (contract number: ICA16-0015), by the Wallenberg AI, Autonomous Systems and Software Program (WASP) funded by the Knut and Alice Wallenberg Foundation, and by ELLIIT.

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

## A EXPERIMENTS

The source code of the experiments and instructions for reproducing the results are available at `https://github.com/devmotion/Calibration_ICLR2021`. Additional material such as automatically generated HTML output and Jupyter notebooks is available at `https://devmotion.github.io/Calibration_ICLR2021/`.

### A.1 ORDINARY LEAST SQUARES

We consider a regression problem with scalar feature $X$ and scalar target $Y$ with input-dependent Gaussian noise that is inspired by a problem by Gustafsson et al. (2020). Feature $X$ is distributed uniformly at random in $[-1, 1]$, and target $Y$ is distributed according to

$$Y \sim \sin(\pi X) + |1 + X|\epsilon,$$

where $\epsilon \sim \mathcal{N}(0, 0.15^2)$. We train a linear regression model $P$ with homoscedastic variance using ordinary least squares and a data set of 100 i.i.d. pairs of feature $X$ and target $Y$ (see Fig. 4).

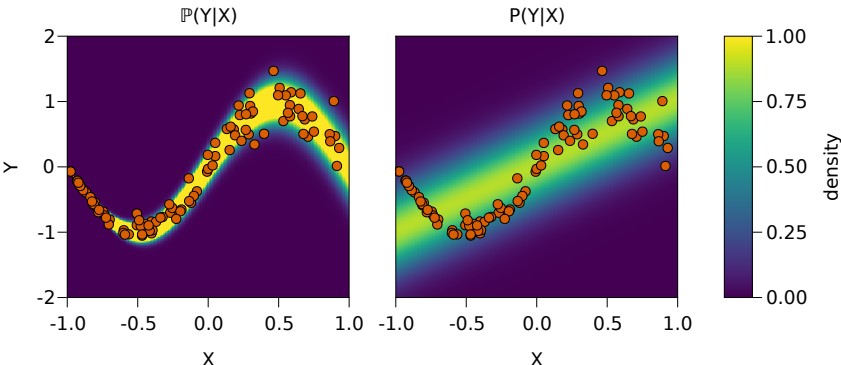

Figure 4: Data generating distribution $\mathbb{P}(Y|X)$ and predicted distribution $P(Y|X)$ of the linear regression model. Training data is indicated by orange dots.

A validation data set of $n = 50$ i.i.d. pairs of $X$ and $Y$ is used to evaluate the empirical cumulative probability

$$n^{-1} \sum_{i=1}^{n} \mathbb{1}_{[0,\tau]} \big( P(Y \leq Y_i | X = X_i) \big)$$

of model $P$ for quantile levels $\tau \in [0, 1]$. Model $P$ would be quantile calibrated (Song et al., 2019) if

$$\tau = \mathbb{P}_{X', Y'} \big( P(Y \leq Y' | X = X') \leq \tau \big)$$

for all $\tau \in [0, 1]$, where $(X, Y)$ and $(X', Y')$ are independent identically distributed pairs of random variables (see Fig. 5).

Additionally, we compute a $p$-value estimate of the null hypothesis $H_0$ that model $P$ is calibrated using an estimation of the quantile of the asymptotic distribution of $n\widehat{\text{SKCE}}_{k,n}$ with 100000 bootstrap samples on the validation data set (see Remark B.2). Kernel $k$ is chosen as the tensor product kernel

$$k\big((p, y), (p', y')\big) = \exp\big(-W_2(p, p')\big) \exp\big(-(y - y')^2/2\big)$$
$$= \exp\left(-\sqrt{(m_p - m_{p'})^2 + (\sigma_p - \sigma_{p'})^2}\right) \exp\big(-(y - y')^2/2\big),$$

where $W_2$ is the 2-Wasserstein distance and $m_p, m_{p'}$ and $\sigma_p, \sigma_{p'}$ denote the mean and the standard deviation of the normal distributions $p$ and $p'$ (see Appendix D.1). We obtain $p < 0.05$ in our experiment, and hence the calibration test rejects $H_0$ at the significance level $\alpha = 0.05$.

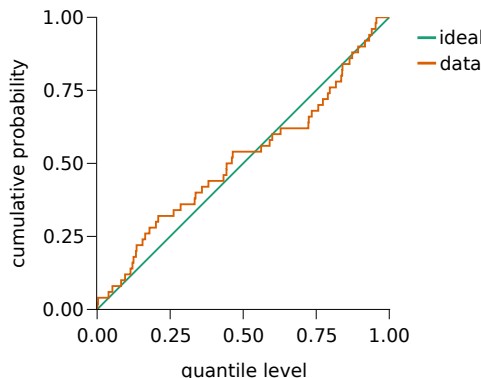

Figure 5: Cumulative probability versus quantile level for the linear regression model on the validation data (orange curve). The green curve indicates the theoretical ideal for a quantile-calibrated model.

## A.2 EMPIRICAL PROPERTIES AND COMPUTATIONAL EFFICIENCY

We study two setups with $d$-dimensional targets $Y$ and normal distributions $P_X$ of the form $\mathcal{N}(c\mathbf{1}_d, 0.1^2\mathbf{I}_d)$ as predictions, where $c \sim \mathrm{U}(0,1)$. Since calibration analysis is only based on the targets and predicted distributions, we neglect features $X$ in these experiments and specify only the distributions of $Y$ and $P_X$.

In the first setup we simulate a calibrated model. We achieve this by sampling targets from the predicted distributions, i.e., by defining the conditional distribution of $Y$ given $P_X$ as

$$Y \,|\, P_X = \mathcal{N}(\mu, \Sigma) \sim \mathcal{N}(\mu, \Sigma).$$

In the second setup we simulate an uncalibrated model of the form

$$Y \,|\, P_X = \mathcal{N}(\mu, \Sigma) \sim \mathcal{N}([0.1, \mu_2, \ldots, \mu_d]^\mathsf{T}, \Sigma).$$

We perform an evaluation of the convergence and computation time of the biased estimator $\widehat{\mathrm{SKCE}}_k$ and the unbiased estimator $\widehat{\mathrm{SKCE}}_{k,B}$ with blocks of size $B \in \{2, \sqrt{n}, n\}$. We use the tensor product kernel

$$
\begin{aligned}
k\big((p,y),(p',y')\big) &= \exp\big(-W_2(p,p')\big) \exp\big(-(y-y')^2/2\big) \\
&= \exp\Big(-\sqrt{(m_p - m_{p'})^2 + (\sigma_p - \sigma_{p'})^2}\Big) \exp\big(-(y-y')^2/2\big),
\end{aligned}
$$

where $W_2$ is the 2-Wasserstein distance and $m_p, m_{p'}$ and $\sigma_p, \sigma_{p'}$ denote the mean and the standard deviation of the normal distributions $p$ and $p'$.

Figures 6 to 9 visualize the mean absolute error and the variance of the resulting estimates for the calibrated and the uncalibrated model with dimensions $d = 1$ and $d = 10$ for 500 independently drawn data sets of $n \in \{4, 16, 64, 256, 1024\}$ samples of $(P_X, Y)$. Computation time indicates the minimum time in the 500 evaluations on a computer with a 3.6 GHz processor. The ground truth values of the uncalibrated models were estimated by averaging the estimates of $\widehat{\mathrm{SKCE}}_{k,1000}$ for 1000 independently drawn data sets of 1000 samples of $(P_X, Y)$ (independent from the data sets used for the evaluation of the estimates). Figures 6 and 7 illustrate that the computational efficiency of $\widehat{\mathrm{SKCE}}_{k,2}$ in comparison with the other estimators comes at the cost of increased error and variance for the calibrated models for fixed numbers of samples.

We compare calibration tests based on the (tractable) asymptotic distribution of $\sqrt{\lfloor n/B \rfloor}\widehat{\mathrm{SKCE}}_{k,B}$ with fixed block size $B \in \{2, \sqrt{n}\}$ (see Remark B.1), the (intractable) asymptotic distribution of $n\widehat{\mathrm{SKCE}}_{k,n}$ which is approximated with 1000 bootstrap samples (see Remark B.2), and a Hotelling's

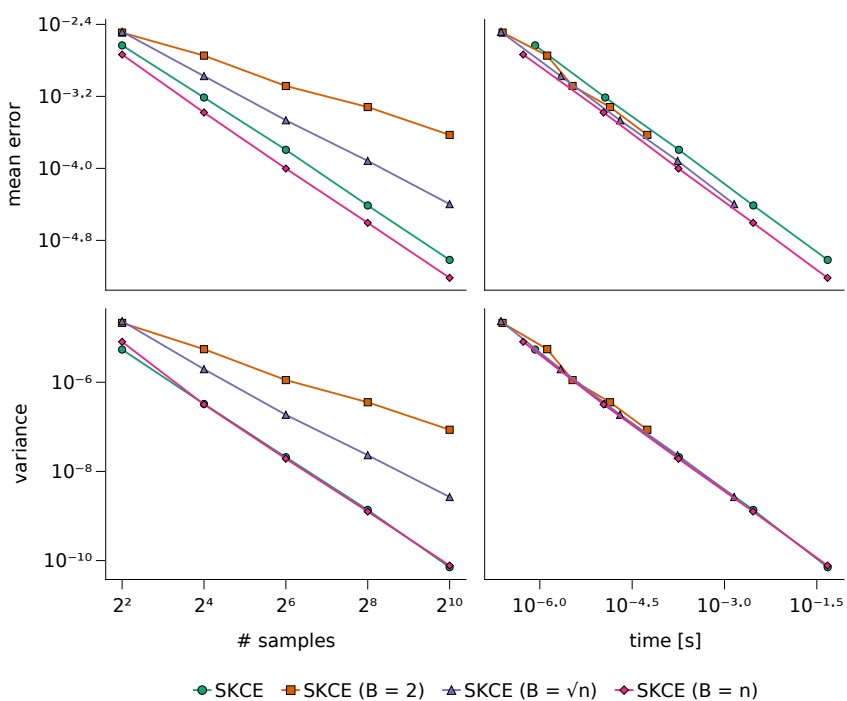

Figure 6: Mean absolute error and variance of 500 calibration error estimates for data sets of $n \in \{4, 16, 64, 256, 1024\}$ samples from the calibrated model of dimension $d = 1$.

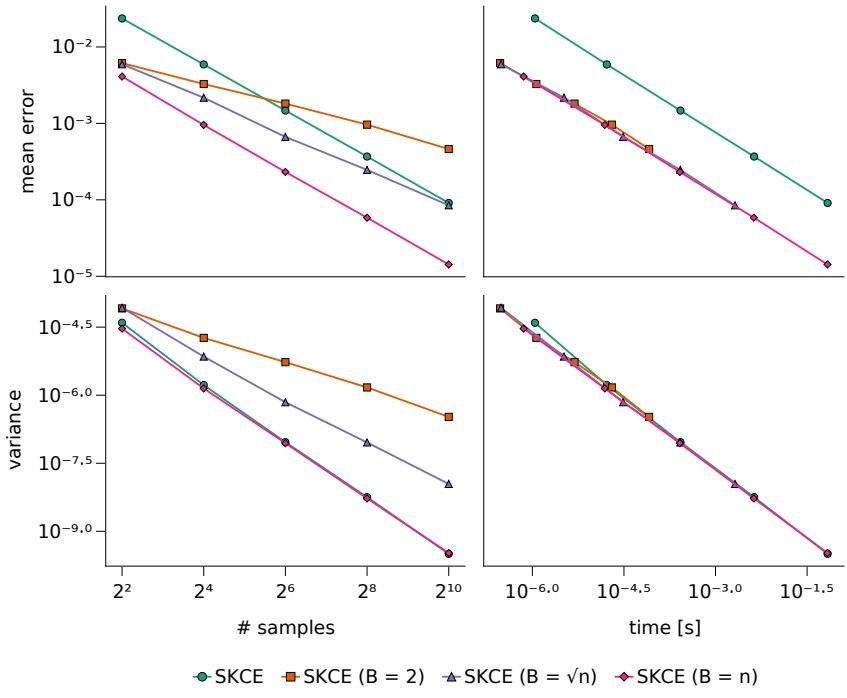

Figure 7: Mean absolute error and variance of 500 calibration error estimates for data sets of $n \in \{4, 16, 64, 256, 1024\}$ samples from the calibrated model of dimension $d = 10$.

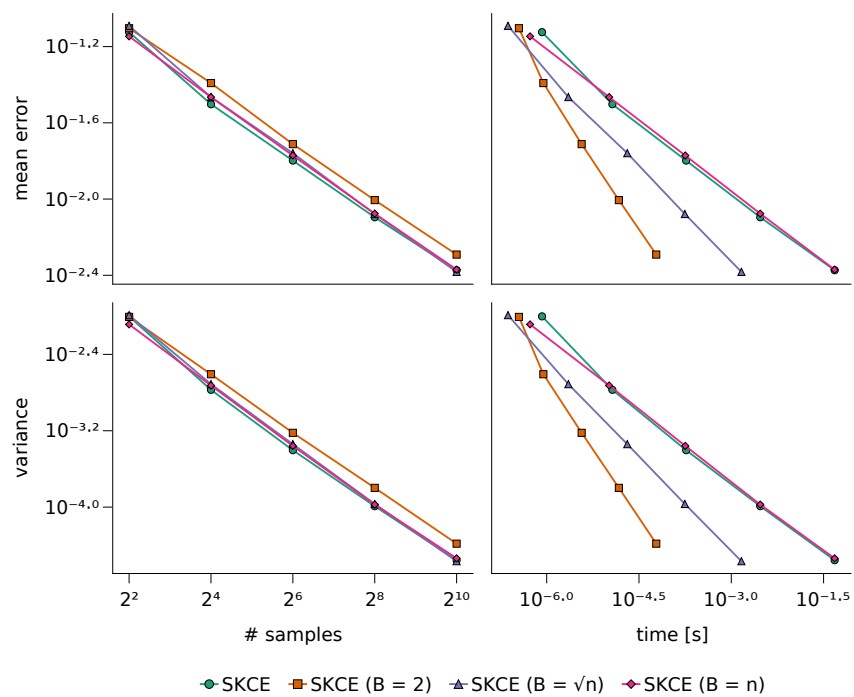

Figure 8: Mean absolute error and variance of 500 calibration error estimates for data sets of $n \in \{4, 16, 64, 256, 1024\}$ samples from the uncalibrated model of dimension $d = 1$.

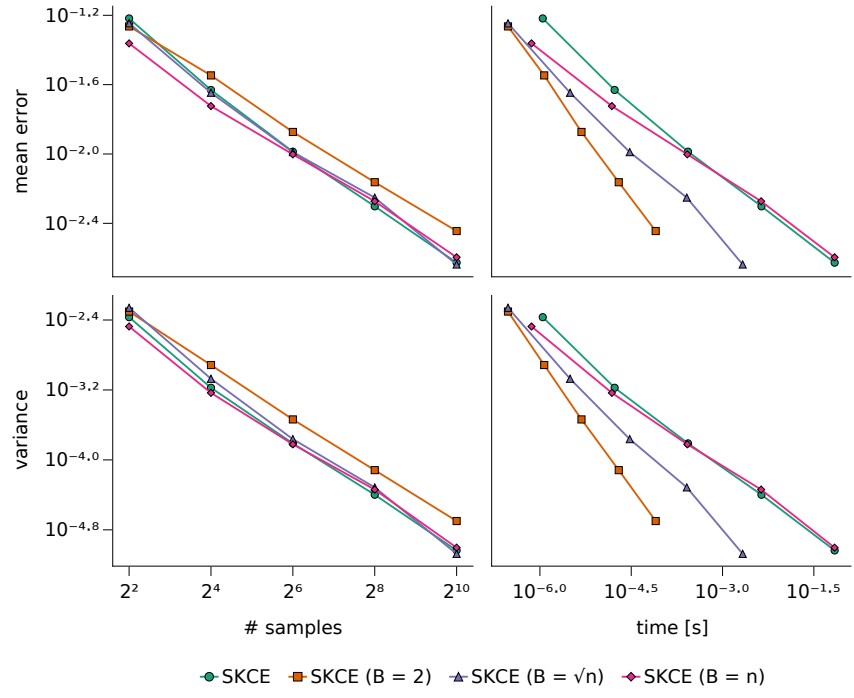

Figure 9: Mean absolute error and variance of 500 calibration error estimates for data sets of $n \in \{4, 16, 64, 256, 1024\}$ samples from the uncalibrated model of dimension $d = 10$.

$T^2$-statistic for $\text{UCME}_{k,10}$ with 10 test locations (see Appendix C). We compute the empirical test errors (percentage of false rejections of the null hypothesis $H_0$ that model $P$ is calibrated if $P$ is calibrated, and percentage of false non-rejections of $H_0$ if $P$ is not calibrated) at a fixed significance level $\alpha = 0.05$ and the minimal computation time for the calibrated and the uncalibrated model with dimensions $d = 1$ and $d = 10$ for 500 independently drawn data sets of $n \in \{4, 16, 64, 256, 1024\}$ samples of $(P_X, Y)$. The 10 test predictions of the CME test are of the form $\mathcal{N}(m, 0.1^2\mathbf{I}_d)$ where $m$ is distributed uniformly at random in the $d$-dimensional unit hypercube $[0, 1]^d$, the corresponding 10 test targets are i.i.d. according to $\mathcal{N}(\mathbf{0}, 0.1^2\mathbf{I}_d)$.

Figures 10 and 11 show that all tests adhere to the set significance level asymptotically as the number of samples increases. The convergence of the CME test with 10 test locations is found to be much slower than the convergence of all other tests. The tests based on the tractable asymptotic distribution of $\sqrt{\lfloor n/B \rfloor}\widehat{\text{SKCE}}_{k,B}$ for fixed block size $B$ are orders of magnitudes faster than the test based on the intractable asymptotic distribution of $n\widehat{\text{SKCE}}_{k,n}$, approximated with 1000 bootstrap samples. We see that the efficiency gain comes at the cost of decreased test power for smaller number of samples, explained by the increasing variance of $\widehat{\text{SKCE}}_{k,B}$ for decreasing block sizes $B$. However, in our examples the test based on $\widehat{\text{SKCE}}_{k,\sqrt{n}}$ still achieves good test power for reasonably large number of samples ($> 30$).

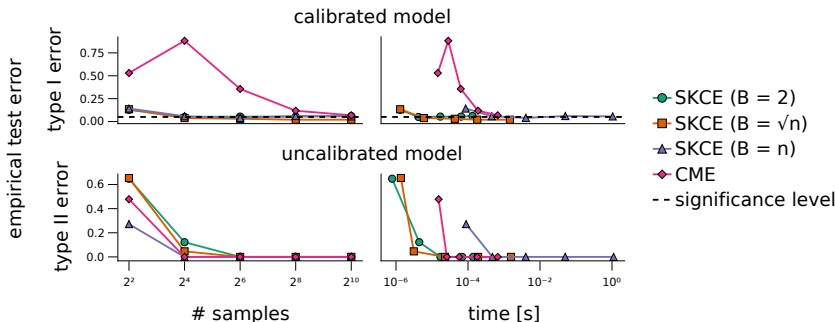

Figure 10: Empirical test errors for 500 data sets of $n \in \{4, 16, 64, 256, 1024\}$ samples from models with targets of dimension $d = 1$. The dashed black line indicates the set significance level $\alpha = 0.05$.

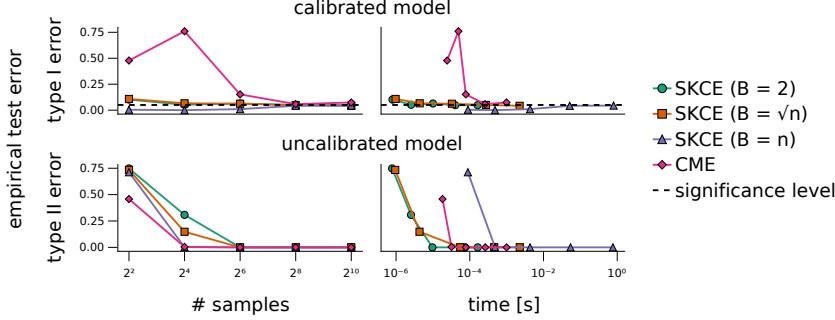

Figure 11: Empirical test errors for 500 data sets of $n \in \{4, 16, 64, 256, 1024\}$ samples from models with targets of dimension $d = 10$. The dashed black line indicates the set significance level $\alpha = 0.05$.

## A.3 FRIEDMAN 1 REGRESSION PROBLEM

We study the so-called Friedman 1 regression problem, which was initially described for 200 inputs in the six-dimensional unit hypercube (Friedman, 1979; Friedman et al., 1983) and later modified to 100 inputs in the 10-dimensional unit hypercube (Friedman, 1991). In this regression problem real-valued target $Y$ depends on input $X$ via

$$Y = 10 \sin(\pi X_1 X_2) + 20(X_3 - 0.5)^2 + 10X_4 + 5X_5 + \epsilon,$$

where noise $\epsilon$ is typically chosen to be independently standard normally distributed. We generate a training data set of 100 inputs distributed uniformly at random in the 10-dimensional unit hypercube and corresponding targets with identically and independently distributed noise following a standard normal distribution.

We consider models $P^{(\theta,\sigma^2)}$ of normal distributions with fixed variance $\sigma^2$

$$P_x^{(\theta,\sigma^2)} = \mathcal{N}(f_\theta(x), \sigma^2),$$

where $f_\theta(x)$, the model of the mean of the distribution $\mathbb{P}(Y|X = x)$, is given by a fully connected neural network with two hidden layers with 200 and 50 hidden units and ReLU activation functions. The parameters of the neural network are denoted by $\theta$.

We use a maximum likelihood approach and train the parameters $\theta$ of the model for 5000 iterations by minimizing the mean squared error on the training data set using ADAM (Kingma & Ba, 2015) (default settings in the machine learning framework Flux.jl (Innes, 2018; Innes et al., 2018)). In each iteration, the variance $\sigma^2$ is set to the maximizer of the likelihood of the training data set.

We train 10 models with different initializations of parameters $\theta$. The initial values of the weight matrices of the neural networks are sampled from the uniform Glorot initialization (Glorot & Bengio, 2010) and the offset vectors are initialized with zeros. In Fig. 12, we visualize estimates of accuracy and calibration measures on the training and test data set with 100 and 50 samples, respectively, for 5000 training iterations. The pinball loss is a common measure and training objective for calibration of quantiles (Song et al., 2019). It is defined as

$$\mathbb{E}_{X,Y} L_\tau\big(Y, \text{quantile}(P_X, \tau)\big),$$

where $L_\tau(y, \tilde{y}) = (1 - \tau)(\tilde{y} - y)_+ + \tau(y - \tilde{y})_+$ and $\text{quantile}(P_x, \tau) = \inf_y\{P_x(Y \leq y) \geq \tau\}$ for quantile level $\tau \in [0, 1]$. In Fig. 12 we plot the average pinball loss (pinball) for quantile levels $\tau \in \{0.05, 0.1, \ldots, 0.95\}$. We evaluate $\widehat{\text{SKCE}}_{k,n}$ (SKCE (unbiased)) and $\widehat{\text{SKCE}}_k$ (SKCE (biased)) for the tensor product kernel

$$k\big((p, y), (p', y')\big) = \exp\big(- W_2(p, p')\big) \exp\big(- (y - y')^2/2\big)$$
$$= \exp\left(- \sqrt{(m_p - m_{p'})^2 + (\sigma_p - \sigma_{p'})^2}\right) \exp\big(- (y - y')^2/2\big),$$

where $W_2$ is the 2-Wasserstein distance and $m_p, m_{p'}$ and $\sigma_p, \sigma_{p'}$ denote the mean and the standard deviation of the normal distributions $p$ and $p'$ (see Appendix D.1). The $p$-value estimate ($p$-value) is computed by estimating the quantile of the asymptotic distribution of $n\widehat{\text{SKCE}}_{k,n}$ with 1000 bootstrap samples (see Remark B.2). The estimates of the mean squared error and the average negative log-likelihood are denoted by MSE and NLL. All estimators indicate consistently that the trained models suffer from overfitting after around 1000 training iterations.

Additionally, we form ensembles of the ten individual models at every training iteration. The evaluations for the ensembles are visualized in Fig. 12 as well. Apart from the unbiased estimates of $\text{SKCE}_k$, the estimates of the ensembles are consistently better than the average estimates of the ensemble members. For the mean squared error and the negative log-likelihood this behaviour is guaranteed theoretically by the generalized mean inequality.

# B THEORY

## B.1 GENERAL SETTING

Let $(\Omega, \mathcal{A}, \mathbb{P})$ be a probability space. Define the random variables $X \colon (\Omega, \mathcal{A}) \to (\mathcal{X}, \Sigma_X)$ and $Y \colon (\Omega, \mathcal{A}) \to (\mathcal{Y}, \Sigma_Y)$ such that $\Sigma_X$ contains all singletons, and denote a version of the regular conditional distribution of $Y$ given $X = x$ by $\mathbb{P}(Y|X = x)$ for all $x \in \mathcal{X}$.

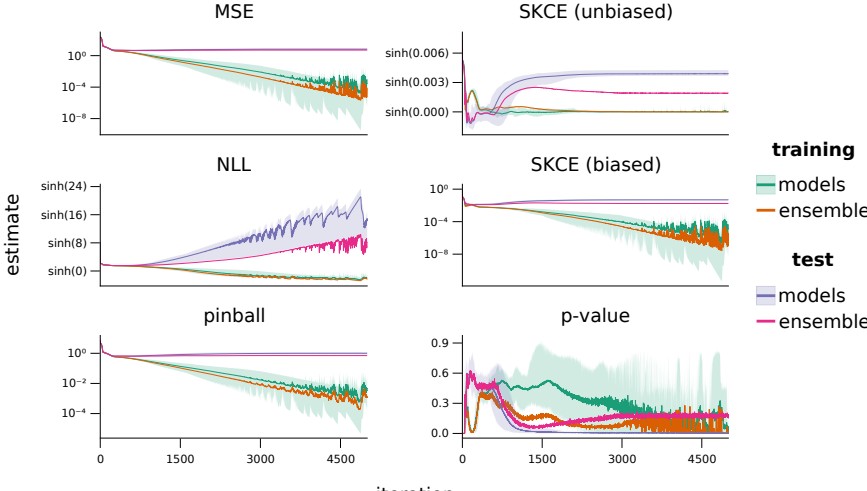

Figure 12: Estimates of different accuracy and calibration measures of ten Gaussian predictive models for the Friedman 1 regression problem versus the number of training iterations. Evaluations on the training data set (100 samples) are displayed in green and orange, and on the test data set (50 samples) in blue and purple. The green and blue line and their surrounding bands represent the mean and the range of the evaluations of the ten models. The orange and purple lines visualize the evaluations of their ensemble.

Let $P \colon (\mathcal{X}, \Sigma_X) \to (\mathcal{P}, \mathcal{B}(\mathcal{P}))$ be a measurable function that maps features in $\mathcal{X}$ to probability measures in $\mathcal{P}$ on the target space $\mathcal{Y}$. We call $P$ a probabilistic model, and denote by $P_x \coloneqq P(x)$ its output for feature $x \in \mathcal{X}$. This gives rise to the random variable $P_X \colon (\Omega, \mathcal{A}) \to (\mathcal{P}, \mathcal{B}(\mathcal{P}))$ as $P_X \coloneqq P(X)$. We denote a version of the regular conditional distribution of $Y$ given $P_X = P_x$ by $\mathbb{P}(Y | P_X = P_x)$ for all $P_x \in \mathcal{P}$.

## B.2   EXPECTED AND MAXIMUM CALIBRATION ERROR

The common definition of the expected and maximum calibration error (Guo et al., 2017; Kull et al., 2019; Naeini et al., 2015; Vaicenavicius et al., 2019) for classification models can be generalized to arbitrary predictive models.

**Definition B.1.** Let $d(\cdot, \cdot)$ be a distance measure of probability distributions of target $Y$, and let $\mu$ be the law of $P_X$. Then we call

$$\mathrm{ECE}_d = \mathbb{E}\, d\big(\mathbb{P}(Y | P_X), P_X\big) \qquad \text{and} \qquad \mathrm{MCE}_d = \mu\text{-}\operatorname{ess\,sup} d\big(\mathbb{P}(Y | P_X), P_X\big)$$

the expected calibration error (ECE) and the maximum calibration error (MCE) of model $P$ with respect to measure $d$, respectively.

## B.3   KERNEL CALIBRATION ERROR

Recall the general notation: Let $k \colon (\mathcal{P} \times \mathcal{Y}) \times (\mathcal{P} \times \mathcal{Y}) \to \mathbb{R}$ be a kernel, amd denote its corresponding RKHS by $\mathcal{H}$.

If not stated otherwise, we assume that

(K1) $k(\cdot, \cdot)$ is Borel-measurable.

(K2) $k$ is integrable with respect to the distributions of $(P_X, Y)$ and $(P_X, Z_X)$, i.e.,

$$\mathbb{E}_{P_X, Y}\, k^{1/2}\big((P_X, Y), (P_X, Y)\big) < \infty$$

and

$$\mathbb{E}_{P_X, Z_X}\, k^{1/2}\big((P_X, Z_X), (P_X, Z_X)\big) < \infty.$$

**Lemma B.1.** *There exist kernel mean embeddings* $\mu_{P_X Y}, \mu_{P_X Z_X} \in \mathcal{H}$ *such that for all* $f \in \mathcal{H}$

$$\langle f, \mu_{P_X Y} \rangle_{\mathcal{H}} = \mathbb{E}_{P_X, Y}\, f(P_X, Y) \qquad and \qquad \langle f, \mu_{P_X Z_X} \rangle_{\mathcal{H}} = \mathbb{E}_{P_X, Z_X}\, f(P_X, Z_X).$$

*This implies that*

$$\mu_{P_X Y} = \mathbb{E}_{P_X, Y}\, k(\cdot, (P_X, Y)) \qquad and \qquad \mu_{P_X Z_X} = \mathbb{E}_{P_X, Z_X}\, k(\cdot, (P_X, Z_X)).$$

*Proof.* The linear operators $T_{P_X Y} f := \mathbb{E}_{P_X, Y}\, f(P_X, Y)$ and $T_{P_X Z_X} f := \mathbb{E}_{P_X, Z_X}\, f(P_X, Z_X)$ for all $f \in \mathcal{H}$ are bounded since

$$|T_{P_X Y} f| = |\mathbb{E}_{P_X, Y}\, f(P_X, Y)| \le \mathbb{E}_{P_X, Y}\, |f(P_X, Y)| = \mathbb{E}_{P_X, Y}\, |\langle k((P_X, Y), \cdot), f \rangle_{\mathcal{H}}|$$
$$\le \mathbb{E}_{P_X, Y}\, \|k((P_X, Y), \cdot)\|_{\mathcal{H}} \|f\|_{\mathcal{H}}] = \|f\|_{\mathcal{H}}\, \mathbb{E}_{P_X, Y}\, k^{1/2}((P_X, Y), (P_X, Y))$$

and similarly

$$|T_{P_X Z_X} f| \le \|f\|_{\mathcal{H}}\, \mathbb{E}_{P_X, Z_X}\, k^{1/2}((P_X, Z_X), (P_X, Z_X)).$$

Thus Riesz representation theorem implies that there exist $\mu_{P_X Y}, \mu_{P_X Z_X} \in \mathcal{H}$ such that $T_{P_X Y} f = \langle f, \mu_{P_X Y} \rangle_{\mathcal{H}}$ and $T_{P_X Z_X} f = \langle f, \mu_{P_X Z_X} \rangle_{\mathcal{H}}$. The reproducing property of $\mathcal{H}$ implies

$$\mu_{P_X Y}(p, y) = \langle k((p, y), \cdot), \mu_{P_X Y} \rangle_{\mathcal{H}} = \mathbb{E}_{P_X, Y}\, k((p, y), (P_X, Y))$$

for all $(p, y) \in \mathcal{P} \times \mathcal{Y}$, and similarly $\mu_{P_X Z_X}(p, y) = \mathbb{E}_{P_X, Z_X}\, k((p, y), (P_X, Z_X))$. $\qquad\square$

**Lemma B.2.** *The squared kernel calibration error (SKCE) with respect to kernel* $k$*, defined as* $\mathrm{SKCE}_k := \mathrm{KCE}_k^2$*, is given by*

$$\mathrm{SKCE}_k = \mathbb{E}_{P_X, Y, P_{X'}, Y'}\, k\big((P_X, Y), (P_{X'}, Y')\big) - 2\, \mathbb{E}_{P_X, Y, P_{X'}, Z_{X'}}\, k\big((P_X, Y), (P_{X'}, Z_{X'})\big)$$
$$+ \mathbb{E}_{P_X, Z_X, P_{X'}, Z_{X'}}\, k\big((P_X, Z_X), (P_{X'}, Z_{X'})\big),$$

*where* $(P_{X'}, Y', Z_{X'})$ *is independently distributed according to the law of* $(P_X, Y, Z_X)$

*Proof.* From Lemma B.1 we know that there exist kernel mean embeddings $\mu_{P_X Y}, \mu_{P_X Z_X} \in \mathcal{H}$ that satisfy

$$\langle f, \mu_{P_X Y} - \mu_{P_X Z_X} \rangle_{\mathcal{H}} = \langle f, \mu_{P_X Y} \rangle_{\mathcal{H}} - \langle f, \mu_{P_X Z_X} \rangle_{\mathcal{H}}$$
$$= \mathbb{E}_{P_X, Y}\, f(P_X, Y) - \mathbb{E}_{P_X, Z_X}\, f(P_X, Z_X)$$

for all $f \in \mathcal{H}$. Hence by the definition of the dual norm

$$\mathrm{CE}_{\mathcal{F}_k} = \sup_{f \in \mathcal{F}_k} \big| \mathbb{E}_{P_X, Y}\, f(P_X, Y) - \mathbb{E}_{P_X, Z_X}\, f(P_X, Z_X) \big|$$
$$= \sup_{f \in \mathcal{F}_k} \big| \langle f, \mu_{P_X, Y} - \mu_{P_X, Z_X} \rangle_{\mathcal{H}} \big| = \|\mu_{P_X, Y} - \mu_{P_X, Z_X}\|_{\mathcal{H}},$$

which implies

$$\mathrm{SKCE}_k = \langle \mu_{P_X Y} - \mu_{P_X Z_X}, \mu_{P_X Y} - \mu_{P_X Z_X} \rangle_{\mathcal{H}}.$$

From Lemma B.1 we obtain

$$\mathrm{SKCE}_k = \mathbb{E}_{P_X, Y, P_{X'}, Y'}\, k\big((P_X, Y), (P_{X'}, Y')\big) - 2\, \mathbb{E}_{P_X, Y, P_{X'}, Z_{X'}}\, k\big((P_X, Y), (P_{X'}, Z'_X)\big)$$
$$+ \mathbb{E}_{P_X, Z_X, P_{X'}, Z'_X}\, k\big((P_X, Z_X), (P_{X'}, Z'_X)\big),$$

which yields the desired result. $\qquad\square$

Recall that $(P_{X_1}, Y_1), \dots, (P_{X_n}, Y_n)$ is a validation data set that is sampled i.i.d. according to the law of $(P_X, Y)$ and that for all $(p, y), (p', y') \in \mathcal{P} \times \mathcal{Y}$

$$h((p, y), (p', y')) := k((p, y), (p', y')) - \mathbb{E}_{Z \sim p}\, k((p, Z), (p', y'))$$
$$- \mathbb{E}_{Z' \sim p'}\, k((p, y), (p', Z')) + \mathbb{E}_{Z \sim p, Z' \sim p'}\, k((p, Z), (p', Z')).$$

**Lemma B.3.** *For all* $i, j = 1, \dots, n$,

$$\big| h\big((P_{X_i}, Y_i), (P_{X_j}, Y_j)\big) \big| < \infty$$

*almost surely.*

*Proof.* Let $i, j \in \{1, \ldots, n\}$. By assumption (K2) we know that

$$\left| k\big((P_{X_i}, Y_i), (P_{X_j}, Y_j)\big) \right| \leq k^{1/2}\big((P_{X_i}, Y_i), (P_{X_i}, Y_i)\big) k^{1/2}\big((P_{X_j}, Y_j), (P_{X_j}, Y_j)\big) < \infty$$

almost surely. Moreover,

$$\begin{aligned}
\left| \mathbb{E}_{Z_{X_i}} k\big((P_{X_i}, Z_{X_i}), (P_{X_j}, Y_j)\big) \right| &\leq \mathbb{E}_{Z_{X_i}} \left| k\big((P_{X_i}, Z_{X_i}), (P_{X_j}, Y_j)\big) \right| \\
&\leq \mathbb{E}_{Z_{X_i}} \left( k^{1/2}\big((P_{X_i}, Z_{X_i}), (P_{X_i}, Z_{X_i})\big) k^{1/2}\big((P_{X_j}, Y_j), (P_{X_j}, Y_j)\big) \right) < \infty
\end{aligned}$$

almost surely, and similarly $\left| \mathbb{E}_{Z_{X_i}, Z_{X_j}} k\big((P_{X_i}, Z_{X_i}), (P_{X_j}, Z_{X_j})\big) \right| < \infty$ almost surely. Thus

$$\begin{aligned}
\left| h\big((P_{X_i}, Y_i), (P_{X_j}, Y_j)\big) \right| &\leq \left| k\big((P_{X_i}, Y_i), (P_{X_j}, Y_j)\big) \right| + \left| \mathbb{E}_{Z_{X_i}} k\big((P_{X_i}, Z_{X_i}), (P_{X_j}, Y_j)\big) \right| \\
&\quad + \left| \mathbb{E}_{Z_{X_j}} k\big((P_{X_i}, Y_i), (P_{X_j}, Z_{X_j})\big) \right| + \left| \mathbb{E}_{Z_{X_i}, Z_{X_j}} k\big((P_{X_i}, Z_{X_i}), (P_{X_j}, Z_{X_j})\big) \right| < \infty
\end{aligned}$$

almost surely. $\qquad\square$

**Lemma 1.** *The plug-in estimator of* $\mathrm{SKCE}_k$ *is non-negatively biased. It is given by*

$$\widehat{\mathrm{SKCE}}_k = \frac{1}{n^2} \sum_{i,j=1}^{n} h\big((P_{X_i}, Y_i), (P_{X_j}, Y_j)\big).$$

*Proof.* From Lemma B.2 we know that $\mathrm{KCE}_k < \infty$, and Lemma B.3 implies that $\widehat{\mathrm{SKCE}}_k < \infty$ almost surely.

For $i = 1, \ldots, n$, the linear operators $T_i f := \mathbb{E}_{Z_{X_i}} f(P_{X_i}, Z_{X_i})$ for $f \in \mathcal{H}$ are bounded almost surely since

$$\begin{aligned}
|T_i f| = \left| \mathbb{E}_{Z_{X_i}} f(P_{X_i}, Z_{X_i}) \right| &\leq \mathbb{E}_{Z_{X_i}} \left| f(P_{X_i}, Z_{X_i}) \right| = \mathbb{E}_{Z_{X_i}} \left| \langle k\big((P_{X_i}, Z_{X_i}), \cdot\big), f \rangle_{\mathcal{H}} \right| \\
&\leq \mathbb{E}_{Z_{X_i}} \left( \left\| k\big((P_{X_i}, Z_{X_i}), \cdot\big) \right\|_{\mathcal{H}} \|f\|_{\mathcal{H}} \right) = \|f\|_{\mathcal{H}} \, \mathbb{E}_{Z_{X_i}} k^{1/2}\big((P_{X_i}, Z_{X_i}), (P_{X_i}, Z_{X_i})\big).
\end{aligned}$$

Hence Riesz representation theorem implies that there exist $\rho_i \in \mathcal{H}$ such that $T_i f = \langle f, \rho_i \rangle_{\mathcal{H}}$ almost surely. From the reproducing property of $\mathcal{H}$ we deduce that $\rho_i(p, y) = \langle k\big((p, y), \cdot\big), \rho_i \rangle_{\mathcal{H}} = \mathbb{E}_{Z_{X_i}} k\big((p, y), (P_{X_i}, Z_{X_i})\big)$ for all $(p, y) \in \mathcal{P} \times \mathcal{Y}$ almost surely.

Thus by the definition of the dual norm the plug-in estimator $\widehat{\mathrm{KCE}}_k$ satisfies

$$\begin{aligned}
\widehat{\mathrm{KCE}}_k &= \sup_{f \in \mathcal{F}_k} \frac{1}{n} \left| \sum_{i=1}^{n} \big( f(P_{X_i}, Y_i) - \mathbb{E}_{Z_{X_i}} f(P_{X_i}, Z_{X_i}) \big) \right| \\
&= \sup_{f \in \mathcal{F}_k} \frac{1}{n} \left| \sum_{i=1}^{n} \langle k\big((P_{X_i}, Y_i), \cdot\big) - \rho_i, f \rangle_{\mathcal{H}} \right| \\
&= \sup_{f \in \mathcal{F}_k} \frac{1}{n} \left| \left\langle \sum_{i=1}^{n} \big( k\big((P_{X_i}, Y_i), \cdot\big) - \rho_i \big), f \right\rangle_{\mathcal{H}} \right| \\
&= \frac{1}{n} \left\| \sum_{i=1}^{n} \big( k\big((G_i, Y_i), \cdot\big) - \rho_i \big) \right\|_{\mathcal{H}} \\
&= \frac{1}{n} \left( \left\langle \sum_{i=1}^{n} k\big((P_{X_i}, Y_i), \cdot\big) - \rho_i, \sum_{i=1}^{n} k\big((P_{X_i}, Y_i), \cdot\big) - \rho_i \right\rangle_{\mathcal{H}} \right)^{1/2} \\
&= \frac{1}{n} \left( \sum_{i,j=1}^{n} h\big((P_{X_i}, Y_i), (P_{X_j}, Y_j)\big) \right)^{1/2} = \widehat{\mathrm{SKCE}}_k^{1/2} < \infty
\end{aligned}$$

almost surely, and hence indeed $\widehat{\mathrm{SKCE}}_k^{1/2}$ is the plug-in estimator of $\mathrm{KCE}_k$.

Since $(P_X, Y), (P_{X'}, Y'), (P_{X_1}, Y_1), \ldots, (P_{X_n}, Y_n)$ are identically distributed and pairwise independent, we obtain

$$
\begin{aligned}
n^2 \mathbb{E}\, \widehat{\mathrm{SKCE}}_k &= \sum_{\substack{i,j=1, \\ i \neq j}}^{n} \mathbb{E}_{P_{X_i}, Y_i, P_{X_j}, Y_j}\, h\big((P_{X_i}, Y_i), (P_{X_j}, Y_j)\big) \\
&\quad + \sum_{i=1}^{n} \mathbb{E}_{P_{X_i}, Y_i}\, h\big((P_{X_i}, Y_i), (P_{X_i}, Y_i)\big) \\
&= n(n-1)\, \mathbb{E}_{P_X, Y, P_{X'}, Y'}\, h\big((P_X, Y), (P_{X'}, Y')\big) + n\, \mathbb{E}_{P_X, Y}\, h\big((P_X, Y), (P_X, Y)\big) \\
&= n(n-1)\mathrm{SKCE}_k + n\, \mathbb{E}_{P_X, Y}\, h\big((P_X, Y), (P_X, Y)\big).
\end{aligned}
$$

$$(B.1)$$

With the same reasoning as above, there exist $\rho, \rho' \in \mathcal{H}$ such that for all $f \in \mathcal{H}$ $\mathbb{E}_{Z_X}\, f(P_X, Z_X) = \langle f, \rho \rangle_{\mathcal{H}}$ and $\mathbb{E}_{Z_{X'}}\, f(P_{X'}, Z_{X'}) = \langle f, \rho' \rangle_{\mathcal{H}}$ almost surely. Thus we obtain

$$
h\big((P_X, Y), (P_{X'}, Y')\big) = \langle k\big((P_X, Y), \cdot\big) - \rho, k\big((P_{X'}, Y'), \cdot\big) - \rho' \rangle_{\mathcal{H}}
$$

almost surely, and therefore by Lemma B.2 and the Cauchy-Schwarz inequality

$$
\begin{aligned}
\mathrm{SKCE}_k &= \mathbb{E}_{P_X, Y, P_{X'}, Y'}\, h\big((P_X, Y), (P_{X'}, Y')\big) \\
&= \mathbb{E}_{P_X, Y, P_{X'}, Y'} \langle k\big((P_X, Y), \cdot\big) - \rho, k\big((G', Y'), \cdot\big) - \rho' \rangle_{\mathcal{H}} \\
&\leq \mathbb{E}_{P_X, Y, P_{X'}, Y'} \left| \langle k\big((P_X, Y), \cdot\big) - \rho, k\big((P_{X'}, Y'), \cdot\big) - \rho' \rangle_{\mathcal{H}} \right| \\
&\leq \mathbb{E}_{P_X, Y, P_{X'}, Y'} \left\| k\big((P_X, Y), \cdot\big) - \rho \right\|_{\mathcal{H}} \left\| k\big((P_{X'}, Y'), \cdot\big) - \rho' \right\|_{\mathcal{H}} \\
&\leq \mathbb{E}_{P_X, Y}^{1/2} \left\| k\big((P_X, Y), \cdot\big) - \rho \right\|_{\mathcal{H}}^2 \mathbb{E}_{P_{X'}, Y'}^{1/2} \left\| k\big((P_{X'}, Y'), \cdot\big) - \rho' \right\|_{\mathcal{H}}^2.
\end{aligned}
$$

Since $(P_X, Y)$ and $(P_{X'}, Y')$ are identically distributed, we obtain

$$
\mathrm{SKCE}_k \leq \mathbb{E}_{P_X, Y} \left\| k\big((P_X, Y), \cdot\big) - \rho \right\|_{\mathcal{H}}^2 = \mathbb{E}_{P_X, Y}\, h\big((P_X, Y), (P_X, Y)\big).
$$

Thus together with Eq. (B.1) we get

$$
n^2 \mathbb{E}\, \widehat{\mathrm{SKCE}}_k \geq n(n-1)\mathrm{SKCE}_k + n\mathrm{SKCE}_k = n^2\mathrm{SKCE}_k,
$$

and hence $\widehat{\mathrm{SKCE}}_k$ has a non-negative bias. $\qquad\square$

**Lemma 2.** *The block estimator of* $\mathrm{SKCE}_k$ *with block size* $B \in \{2, \ldots, n\}$, *given by*

$$
\widehat{\mathrm{SKCE}}_{k,B} \coloneqq \left\lfloor \frac{n}{B} \right\rfloor^{-1} \sum_{b=1}^{\lfloor n/B \rfloor} \binom{B}{2}^{-1} \sum_{(b-1)B < i < j \leq bB} h\big((P_{X_i}, Y_i), (P_{X_j}, Y_j)\big),
$$

*is an unbiased estimator of* $\mathrm{SKCE}_k$.

*Proof.* From Lemma B.2 we know that $\mathrm{SKCE}_k < \infty$, and Lemma B.3 implies that $\widehat{\mathrm{SKCE}}_{k,B} < \infty$ almost surely.

For $b \in \{1, \ldots, \lfloor n/B \rfloor\}$, let

$$
\widehat{\eta}_b \coloneqq \binom{B}{2}^{-1} \sum_{(b-1)B < i < j \leq bB} h\big((P_{X_i}, Y_i), (P_{X_j}, Y_j)\big) \tag{B.2}
$$

be the estimator of the $b$th block. From Lemma B.3 it follows that $\widehat{\eta}_b < \infty$ almost surely for all $b$. Moreover, for all $b$, $\widehat{\eta}_b$ is a so-called U-statistic of $\mathrm{SKCE}_k$ and hence satisfies $\mathbb{E}\, \widehat{\eta}_b = \mathrm{SKCE}_k$ (see, e.g., van der Vaart, 1998). Since $(P_{X_1}, Y_1), \ldots, (P_{X_n}, Y_n)$ are pairwise independent, this implies that $\widehat{\mathrm{SKCE}}_{k,B}$ is an unbiased estimator of $\mathrm{SKCE}_k$. $\qquad\square$

## B.4 CALIBRATION TESTS

**Lemma B.4.** *Let $B \in \{2, \ldots, n\}$. If $\mathbb{V}_{P_X, Y, P_{X'}, Y'} h\big((P_X, Y), (P_{X'}, Y')\big) < \infty$, then for all $b \in \{1, \ldots, \lfloor n/B \rfloor\}$*

$$\mathbb{V} \widehat{\eta}_b = \sigma_B^2 := \binom{B}{2}^{-1} \Big( 2(B-2)\zeta_1 + \mathbb{V}_{P_X, Y, P_{X'}, Y'} h\big((P_X, Y), (P_{X'}, Y')\big) \Big),$$

*where $\widehat{\eta}_b$ is defined according to Eq.* (B.2) *and*

$$\zeta_1 := \mathbb{E}_{P_X, Y} \mathbb{E}^2_{P_{X'}, Y'} h\big((P_X, Y), (P_{X'}, Y')\big) - \mathrm{SKCE}_k^2. \tag{B.3}$$

*If model $P$ is calibrated, it simplifies to*

$$\sigma_B^2 = \binom{B}{2}^{-1} \mathbb{E}_{P_X, Y, P_{X'}, Y'} h^2\big((P_X, Y), (P_{X'}, Y')\big).$$

*Proof.* Let $b \in \{1, \ldots, \lfloor n/B \rfloor\}$. Since $\mathbb{V}_{P_X, Y, P_{X'}, Y'} h\big((P_X, Y), (P_{X'}, Y')\big) < \infty$, the Cauchy-Schwarz inequality implies $\mathbb{V} \widehat{\eta}_b < \infty$ as well.

As mentioned in the proof of Lemma 2 above, $\widehat{\eta}_b$ is a U-statistic of $\mathrm{SKCE}_k$. From the general formula of the variance of a U-statistic (see, e.g., Hoeffding, 1948, p. 298–299) we obtain

$$\mathbb{V} \widehat{\eta}_b = \binom{B}{2}^{-1} \left( \binom{2}{1}\binom{B-2}{2-1} \zeta_1 + \binom{2}{2}\binom{B-2}{2-2} \mathbb{V}_{P_X, Y, P_{X'}, Y'} h\big((P_X, Y), (P_{X'}, Y')\big) \right)$$

$$= \binom{B}{2}^{-1} \Big( 2(B-2)\zeta_1 + \mathbb{V}_{P_X, Y, P_{X'}, Y'} h\big((P_X, Y), (P_{X'}, Y')\big) \Big),$$

where

$$\zeta_1 = \mathbb{E}_{P_X, Y} \mathbb{E}^2_{P_{X'}, Y'} h\big((P_X, Y), (P_{X'}, Y')\big) - \mathrm{SKCE}_k^2.$$

If model $P$ is calibrated, then $(P_X, Y) \stackrel{d}{=} (P_X, Z)$, and hence for all $(p, y) \in \mathcal{P} \times \mathcal{Y}$

$$\mathbb{E}_{P_X, Y} h\big((p, y), (P_X, Y)\big) = \mathbb{E}_{P_X, Y} k\big((p, y), (P_X, Y)\big) - \mathbb{E}_{Z' \sim p} \mathbb{E}_{P_X, Y} k\big((p, Z'), (P_X, Y)\big)$$
$$- \mathbb{E}_{P_X, Z} k\big((p, y), (P_X, Z)\big) + \mathbb{E}_{Z' \sim p} \mathbb{E}_{P_X, Z} k\big((p, Z'), (P_X, Y)\big)$$
$$= 0.$$

This implies $\zeta_1 = \mathbb{E}_{P_X, Y} \mathbb{E}^2_{P_{X'}, Y'} h\big((P_X, Y), (P_{X'}, Y')\big) = 0$ and $\mathrm{SKCE}_k^2 = 0$ due to Lemma B.2. Thus

$$\sigma_B^2 = \binom{B}{2}^{-1} \mathbb{E}_{P_X, Y, P_{X'}, Y'} h^2\big((P_X, Y), (P_{X'}, Y')\big),$$

as stated above. $\qquad \square$

**Corollary B.1.** *Let $B \in \{2, \ldots, n\}$. If $\mathbb{V}_{P_X, Y, P_{X'}, Y'} h\big((P_X, Y), (P_{X'}, Y')\big) < \infty$, then*

$$\mathbb{V} \widehat{SKCE}_{k, B} = \lfloor n/B \rfloor^{-1} \sigma_B^2.$$

*where $\sigma_B^2$ is defined according to Lemma B.4.*

*Proof.* Since the estimators $\widehat{\eta}_1, \ldots, \widehat{\eta}_{\lfloor n/B \rfloor}$ in each block are pairwise independent, this is an immediate consequence of Lemma B.4. $\qquad \square$

**Corollary B.2.** *Let $B \in \{2, \ldots, n\}$. If $\mathbb{V}_{P_X, Y, P_{X'}, Y'} h\big((P_X, Y), (P_{X'}, Y')\big) < \infty$, then*

$$\sqrt{\lfloor n/B \rfloor} \big(\widehat{SKCE}_{k, B} - \mathrm{SKCE}_k\big) \stackrel{d}{\to} \mathcal{N}\big(0, \sigma_B^2\big) \qquad \text{as } n \to \infty,$$

*where block size $B$ is fixed and $\sigma_B^2$ is defined according to Lemma B.4.*

*Proof.* The result follows from Lemma 2, Lemma B.4, and the central limit theorem (see, e.g., Serfling, 1980, Theorem A in Section 1.9). $\qquad \square$

*Remark* B.1. Corollary B.2 shows that $\widehat{\mathrm{SKCE}}_{k,B}$ is a consistent estimator of $\mathrm{SKCE}_k$ in the large sample limit as $n \to \infty$ with fixed number $B$ of samples per block. In particular, for the linear estimator with $B = 2$ we obtain

$$\sqrt{\lfloor n/2 \rfloor}\big(\widehat{\mathrm{SKCE}}_{k,2} - \mathrm{SKCE}_k\big) \xrightarrow{d} \mathcal{N}\big(0, \sigma_2^2\big) \qquad \text{as } n \to \infty.$$

Moreover, Lemma B.4 and Corollary B.2 show that the $p$-value of the null hypothesis that model $P$ is calibrated can be estimated by

$$\Phi\bigg( - \frac{\sqrt{\lfloor n/B \rfloor}\widehat{\mathrm{SKCE}}_{k,B}}{\widehat{\sigma}_B}\bigg),$$

where $\Phi$ is the cumulative distribution function of the standard normal distribution and $\widehat{\sigma}_B$ is the empirical standard deviation of the block estimates $\widehat{\eta}_1, \ldots, \widehat{\eta}_{\lfloor n/B \rfloor}$, and

$$\Phi\bigg( - \frac{\sqrt{\lfloor n/B \rfloor B(B-1)}\widehat{\mathrm{SKCE}}_{k,B}}{\sqrt{2}\widehat{\sigma}}\bigg),$$

where $\widehat{\sigma}^2$ is an estimate of $\mathbb{E}_{P_X,Y,P_{X'},Y'} h^2\big((P_X, Y), (P_{X'}, Y')\big)$. Similar $p$-value approximations for the two-sample test with blocks of fixed size were used by Chwialkowski et al. (2015).

**Corollary B.3.** *Assume* $\mathbb{V}_{P_X,Y,P_{X'},Y'} h\big((P_X, Y), (P_{X'}, Y')\big) < \infty$. *Let* $s \in \{1, \ldots, \lfloor n/2 \rfloor\}$. *Then for all* $b \in \{1, \ldots, s\}$

$$\sqrt{B}\big(\widehat{\eta}_b - \mathrm{SKCE}_k\big) \xrightarrow{d} \mathcal{N}(0, 4\zeta_1) \qquad \text{as } B \to \infty, \tag{B.4}$$

*where* $\widehat{\eta}_b$ *is defined according to Eq.* (B.2) *with* $n = Bs$, *the number* $s$ *of equally-sized blocks is fixed, and* $\zeta_1$ *is defined according to Eq.* (B.3).

*If model $P$ is calibrated, then* $\sqrt{B}\big(\widehat{\eta}_b - \mathrm{SKCE}_k\big) = \sqrt{B}\widehat{\eta}_b$ *is asymptotically tight since* $\zeta_1 = 0$, *and*

$$B\widehat{\eta}_b \xrightarrow{d} \sum_{i=1}^{\infty} \lambda_i(Z_i - 1) \qquad \text{as } B \to \infty, \tag{B.5}$$

*where* $Z_i$ *are independent* $\chi_1^2$ *distributed random variables and* $\lambda_i \in \mathbb{R}$ *are eigenvalues of the Hilbert-Schmidt integral operator*

$$Kf(p, y) := \mathbb{E}_{P_X,Y}\big(h((p, y), (P_X, Y))f(P_X, Y)\big)$$

*for Borel-measurable functions* $f: \mathcal{P} \times \mathcal{Y} \to \mathbb{R}$ *with* $\mathbb{E}_{P_X,Y} f^2(P_X, Y) < \infty$.

*Proof.* Let $s \in \{1, \ldots, \lfloor n/2 \rfloor\}$ and $b \in \{1, \ldots, s\}$. As mentioned above in the proof of Lemma 2, the estimator $\widehat{\eta}_b$, defined according to Eq. (B.2), is a so-called U-statistic of $\mathrm{SKCE}_k$ (see, e.g., van der Vaart, 1998). Thus Eq. (B.4) follows from the asymptotic behaviour of U-statistics (see, e.g., van der Vaart, 1998, Theorem 12.3).

If $P$ is calibrated, then we know from the proof of Lemma B.4 that $\zeta_1 = 0$, and hence $\widehat{\eta}_b$ is a so-called degenerate- U-statistic (see, e.g., van der Vaart, 1998, Section 12.3). From the theory of degenerate U-statistics it follows that the sequence $B\widehat{\eta}_b$ converges in distribution to the limit distribution in Eq. (B.5), which is known as Gaussian chaos. $\square$

**Corollary B.4.** *Assume* $\mathbb{V}_{P_X,Y,P_{X'},Y'} h\big((P_X, Y), (P_{X'}, Y')\big) < \infty$. *Let* $s \in \{1, \ldots, \lfloor n/2 \rfloor\}$. *Then*

$$\sqrt{B}\big(\widehat{\mathrm{SKCE}}_{k,B} - \mathrm{SKCE}_k\big) \xrightarrow{d} \mathcal{N}(0, 4s^{-1}\zeta_1) \qquad \text{as } B \to \infty,$$

*where the number $s$ of equally-sized blocks is fixed,* $n = Bs$, *and* $\zeta_1$ *is defined according to Eq.* (B.3).

*If model $P$ is calibrated, then* $\sqrt{B}\big(\widehat{\mathrm{SKCE}}_{k,B} - \mathrm{SKCE}_k\big) = \sqrt{B}\widehat{\mathrm{SKCE}}_{k,B}$ *is asymptotically tight since* $\zeta_1 = 0$, *and*

$$B\widehat{\mathrm{SKCE}}_{k,B} \xrightarrow{d} s^{-1}\sum_{i=1}^{\infty} \lambda_i(Z_i - s) \qquad \text{as } B \to \infty,$$

*where* $Z_i$ *are independent* $\chi_s^2$ *distributed random variables and* $\lambda_i \in \mathbb{R}$ *are eigenvalues of the Hilbert-Schmidt integral operator*

$$Kf(p, y) := \mathbb{E}_{P_X,Y}\big(h((p, y), (P_X, Y))f(P_X, Y)\big)$$

*for Borel-measurable functions* $f: \mathcal{P} \times \mathcal{Y} \to \mathbb{R}$ *with* $\mathbb{E}_{P_X,Y} f^2(P_X, Y) < \infty$.

*Proof.* Since the estimators $\widehat{\eta}_1, \ldots, \widehat{\eta}_s$ in each block are pairwise independent, this is an immediate consequence of Corollary B.3. □

*Remark* B.2. Corollary B.4 shows that $\widehat{\text{SKCE}}_{k,B}$ is a consistent estimator of $\text{SKCE}_k$ in the large sample limit as $B \to \infty$ with fixed number $\lfloor n/B \rfloor$ of blocks. Moreover, for the minimum variance unbiased estimator with $B = n$, Corollary B.4 shows that under the null hypothesis that model $P$ is calibrated

$$n\widehat{\text{SKCE}}_{k,n} \xrightarrow{d} \sum_{i=1}^{\infty} \lambda_i (Z_i - 1) \qquad \text{as } n \to \infty,$$

where $Z_i$ are independent $\chi_1^2$ distributed random variables. Unfortunately quantiles of the limit distribution of $\sum_{i=1}^{\infty} \lambda_i (Z_i - 1)$ (and hence the $p$-value of the null hypothesis that model $P$ is calibrated) can not be computed analytically but have to be estimated by, e.g., bootstrapping (Arcones & Giné, 1992), using a Gram matrix spectrum (Gretton et al., 2009), fitting Pearson curves (Gretton et al., 2007), or using a Gamma approximation (Johnson et al., 1994, p. 343, p. 359).

**Corollary B.5.** *Assume* $\mathbb{V}_{P_X,Y,P_{X'},Y'} \, h\big((P_X, Y), (P_{X'}, Y')\big) < \infty$. *Then*

$$\sqrt{\lfloor n/B \rfloor B}\big(\widehat{\text{SKCE}}_{k,B} - \text{SKCE}_k\big) \xrightarrow{d} \mathcal{N}(0, 4\zeta_1) \qquad \text{as } B \to \infty \text{ and } \lfloor n/B \rfloor \to \infty, \qquad \text{(B.6)}$$

*where $B$ is the block size and $s$ is the number of equally-sized blocks, $n = Bs$, and $\zeta_1$ is defined according to Eq.* (B.3).

*If model $P$ is calibrated, then* $\sqrt{\lfloor n/B \rfloor B}\big(\widehat{\text{SKCE}}_{k,B} - \text{SKCE}_k\big) = \sqrt{\lfloor n/B \rfloor B}\widehat{\text{SKCE}}_{k,B}$ *is asymptotically tight since $\zeta_1 = 0$, and*

$$\sqrt{\lfloor n/B \rfloor}B\widehat{\text{SKCE}}_{k,B} \xrightarrow{d} \mathcal{N}\left(0, \sum_{i=1}^{\infty} \lambda_i^2\right) \qquad \text{as } B \to \infty \text{ and } \lfloor n/B \rfloor \to \infty,$$

*where $\lambda_i \in \mathbb{R}$ are eigenvalues of the Hilbert-Schmidt integral operator*

$$Kf(p, y) \coloneqq \mathbb{E}_{P_X,Y}\big(h((p, y), (P_X, Y))f(P_X, Y)\big)$$

*for Borel-measurable functions $f \colon \mathcal{P} \times \mathcal{Y} \to \mathbb{R}$ with $\mathbb{E}_{P_X,Y} f^2(P_X, Y) < \infty$.*

*Proof.* The result follows from Corollary B.3 and the central limit theorem (see, e.g., Serfling, 1980, Theorem A in Section 1.9). □

*Remark* B.3. Corollary B.5 shows that $\widehat{\text{SKCE}}_{k,B}$ is a consistent estimator of $\text{SKCE}_k$ in the large sample limit as $B \to \infty$ and $\lfloor n/B \rfloor \to \infty$, i.e., as both the number of samples per block and the number of blocks go to infinity. Moreover, Corollaries B.3 and B.5 show that the $p$-value of the null hypothesis that $P$ is calibrated can be estimated by

$$\Phi\left(-\frac{\sqrt{\lfloor n/B \rfloor}\widehat{\text{SKCE}}_{k,B}}{\widehat{\sigma}_B}\right),$$

where $\widehat{\sigma}_B$ is the empirical standard deviation of the block estimates $\widehat{\eta}_1, \ldots, \widehat{\eta}_{\lfloor n/B \rfloor}$. Similar $p$-value approximations for the two-sample problem with blocks of increasing size were proposed and applied by Zaremba et al. (2013).

## C  CALIBRATION MEAN EMBEDDING

### C.1  DEFINITION

Similar to the unnormalized mean embedding (UME) proposed by Chwialkowski et al. (2015) in the standard MMD setting, instead of the calibration error $\text{CE}_{\mathcal{F}_k} = \|\mu_{P_X Y} - \mu_{P_X Z_X}\|_{\mathcal{H}}$ we can consider the unnormalized calibration mean embedding (UCME).

**Definition C.1.** Let $J \in \mathbb{N}$. The unnormalized calibration mean embedding (UCME) for kernel $k$ with $J$ test locations is defined as the random variable

$$\mathrm{UCME}^2_{k,J} = J^{-1} \sum_{j=1}^{J} \left( \mu_{P_X Y}(T_j) - \mu_{P_X Z_X}(T_j) \right)^2$$

$$= J^{-1} \sum_{j=1}^{J} \left( \mathbb{E}_{P_X,Y}\, k(T_j, (P_X, Y)) - \mathbb{E}_{P_X, Z_X}\, k(T_j, (P_X, Z_X)) \right)^2,$$

where $T_1, \ldots, T_J$ are i.i.d. random variables (so-called test locations) whose distribution is absolutely continuous with respect to the Lebesgue measure on $\mathcal{P} \times \mathcal{Y}$.

As mentioned above, in many machine learning applications we actually have $\mathcal{P} \times \mathcal{Y} \subset \mathbb{R}^d$ (up to some isomorphism). In such a case, if $k$ is an analytic, integrable, characteristic kernel, then for each $J \in \mathbb{N}$ $\mathrm{UCME}_{k,J}$ is a random metric between the distributions of $(P_X, Y)$ and $(P_X, Z_X)$, as shown by Chwialkowski et al. (2015, Theorem 2). In particular, this implies that $\mathrm{UCME}_{k,J} = 0$ almost surely if and only if the two distributions are equal.

## C.2 ESTIMATION

Again we assume $(P_{X_1}, Y_1), \ldots, (P_{X_n}, Y_n)$ is a validation data set of predictions and targets, which are i.i.d. according to the law of $(P_X, Y)$. The consistent, but biased, plug-in estimator of $\mathrm{UCME}^2_{k,J}$ is given by

$$\widehat{\mathrm{UCME}}^2_{k,J} = J^{-1} \sum_{j=1}^{J} \left( n^{-1} \sum_{i=1}^{n} \left( k\big(T_j, (P_{X_i}, Y_i)\big) - \mathbb{E}_{Z_{X_i}}\, k\big(T_j, (P_{X_i}, Z_{X_i})\big) \right) \right)^2.$$

## C.3 CALIBRATION MEAN EMBEDDING TEST

As Chwialkowski et al. (2015) note, if model $P$ is calibrated, for every fixed sequence of unique test locations $\sqrt{n}\widehat{\mathrm{UCME}}^2_{k,J}$ converges in distribution to a sum of correlated $\chi^2$ random variables, as $n \to \infty$. The estimation of this asymptotic distribution, and its quantiles required for hypothesis testing, requires a bootstrap or permutation procedure, which is computationally expensive. Hence Chwialkowski et al. (2015) proposed the following test based on Hotelling's $T^2$-statistic (Hotelling, 1931).

For $i = 1, \ldots, n$, let

$$Z_i := \begin{pmatrix} k\big(T_1, (P_{X_i}, Y_i)\big) - \mathbb{E}_{Z_{X_i}}\, k\big(T_1, (P_{X_i}, Z_{X_i})\big) \\ \vdots \\ k\big(T_J, (P_{X_i}, Y_i)\big) - \mathbb{E}_{Z_{X_i}}\, k\big(T_J, (P_{X_i}, Z_{X_i})\big) \end{pmatrix} \in \mathbb{R}^J,$$

and denote the empirical mean and covariance matrix of $Z_1, \ldots, Z_n$ by $\overline{Z}$ and $S$, respectively. If $\mathrm{UCME}_{k,J}$ is a random metric between the distributions of $(P_X, Y)$ and $(P_X, Z_X)$, then the test statistic

$$Q_n := n\overline{Z}^T S^{-1} \overline{Z}$$

is almost surely asymptotically $\chi^2$ distributed with $J$ degrees of freedom if model $P$ is calibrated, as $n \to \infty$ with $J$ fixed; moreover, if model $P$ is uncalibrated, then for any fixed $r \in \mathbb{R}$ almost surely $\mathbb{P}(Q_n > r) \to 1$ as $n \to \infty$ (Chwialkowski et al., 2015, Proposition 2). We call the resulting calibration test calibration mean embedding (CME) test.

## D KERNEL CHOICE

A natural choice for the kernel $k \colon (\mathcal{P} \times \mathcal{Y}) \times (\mathcal{P} \times \mathcal{Y}) \to \mathbb{R}$ on the product space of predicted distributions $\mathcal{P}$ and targets $\mathcal{Y}$ is a tensor product kernel of the form $k = k_{\mathcal{P}} \otimes k_{\mathcal{Y}}$, i.e., a kernel of the form

$$k\big((p, y), (p', y')\big) = k_{\mathcal{P}}(p, p') k_{\mathcal{Y}}(y, y'),$$

where $k_{\mathcal{P}} \colon \mathcal{P} \times \mathcal{P} \to \mathbb{R}$ and $k_{\mathcal{Y}} \colon \mathcal{Y} \times \mathcal{Y} \to \mathbb{R}$ are kernels on the spaces of predicted distributions and targets, respectively.

As discussed in Section 3.1, if kernel $k$ is characteristic, then the kernel calibration error $\mathrm{KCE}_k$ of model $P$ is zero if and only if $P$ is calibrated. Unfortunately, as shown by Szabó & Sriperumbudur (2018, Example 1), even if $k_{\mathcal{P}}$ and $k_{\mathcal{Y}}$ are characteristic, the tensor product kernel $k = k_{\mathcal{P}} \otimes k_{\mathcal{Y}}$ might not be characteristic. However, when analyzing calibration, it is sufficient to be able to distinguish distributions for which the conditional distributions $\mathbb{P}(Y|P_X)$ and $\mathbb{P}(Z_X|P_X) = P_X$ are not equal almost surely. Thus it is sufficient if $k_{\mathcal{Y}}$ is characteristic and $k_{\mathcal{P}}$ is non-zero almost surely.

Many common kernels such as the Gaussian and Laplacian kernel on $\mathbb{R}^d$ are characteristic and can therefore be chosen as kernel $k_{\mathcal{Y}}$ for real-valued target spaces. The choice of $k_{\mathcal{P}}$ might be less obvious since $\mathcal{P}$ is a space of probability distributions. Intuitively one might want to use kernels of the form
$$k_{\mathcal{P}}(p, p') = \exp\left(-\lambda d_{\mathcal{P}}^{\nu}(p, p')\right), \tag{D.1}$$
where $d_{\mathcal{P}} \colon \mathcal{P} \times \mathcal{P} \to \mathbb{R}$ is a metric on $\mathcal{P}$ and $\nu, \lambda > 0$ are kernel hyperparameters. Kernels of this form would be a generalization of the Gaussian and Laplacian kernel, and would clearly be non-zero almost surely.

Unfortunately, this construction does not necessarily yield valid kernels. Most prominently, the Wasserstein distance does not lead to valid kernels $k_{\mathcal{P}}$ in general (Peyré & Cuturi, 2019, Chapter 8.3). However, if $d_{\mathcal{P}}(\cdot, \cdot)$ is a Hilbertian metric, i.e., a metric of the form
$$d_{\mathcal{P}}(p, p') = \left\| \phi(p) - \phi(p') \right\|_H$$
for some Hilbert space $H$ and mapping $\phi \colon \mathcal{P} \to H$, then $k_{\mathcal{P}}$ in Eq. (D.1) is a valid kernel for all $\lambda > 0$ and $\nu \in (0, 2]$ (Berg et al., 1984, Corollary 3.3.3, Proposition 3.2.7).

### D.1 Normal distributions

Assume that $\mathcal{Y} = \mathbb{R}^d$ and $\mathcal{P} = \{\mathcal{N}(\mu, \Sigma) \colon \mu \in \mathbb{R}^d, \Sigma \in \mathbb{R}^{d \times d} \text{ psd}\}$, i.e., the model outputs normal distributions $P_X = \mathcal{N}(\mu_X, \Sigma_X)$. The distribution of these outputs is defined by the distribution of their mean $\mu_X$ and covariance matrix $\Sigma_X$.

Let $P_x = \mathcal{N}(\mu_x, \Sigma_x) \in \mathcal{P}$, $y \in \mathcal{Y} = \mathbb{R}^d$, and $\gamma > 0$. We obtain

$$\mathbb{E}_{Z_x \sim P_x} \exp\left(-\gamma \|Z_x - y\|_2^2\right)$$
$$= \left|\mathbf{I}_d + 2\gamma\Sigma_x\right|^{-1/2} \exp\left(-\gamma(\mu_x - y)^{\mathsf{T}}\left(\mathbf{I}_d + 2\gamma\Sigma_x\right)^{-1}(\mu_x - y)\right)$$

from Mathai & Provost (1992, Theorem 3.2.a.3). In particular, if $\Sigma_x = \mathrm{diag}(\Sigma_{x,1}, \ldots, \Sigma_{x,d})$, then

$$\mathbb{E}_{Z_x \sim P_x} \exp\left(-\gamma \|Z_x - y\|_2^2\right)$$
$$= \prod_{i=1}^{d} \left[\left(1 + 2\gamma\Sigma_{x,i}\right)^{-1/2} \exp\left(-\gamma\left(1 + 2\gamma\Sigma_{x,i}\right)^{-1}\left(\mu_{x,i} - y_i\right)^2\right)\right].$$

Let $P_{x'} = \mathcal{N}(\mu_{x'}, \Sigma_{x'})$ be another normal distribution. Then we have

$$\mathbb{E}_{Z_x \sim P_x, Z_{x'} \sim P_{x'}} \exp\left(-\gamma \|Z_x - Z_{x'}\|_2^2\right)$$
$$= \left|\mathbf{I}_d + 2\gamma\Sigma_x\right|^{-1/2} \mathbb{E}_{Z_{x'} \sim P_{x'}} \exp\left(-\gamma(\mu_x - Z_{x'})^{\mathsf{T}}\left(\mathbf{I}_d + 2\gamma\Sigma_x\right)^{-1}(\mu_x - Z_{x'})\right)$$
$$= \left|\mathbf{I}_d + 2\gamma(\Sigma_x + \Sigma_{x'})\right|^{-1/2} \exp\left(-\gamma(\mu_x - \mu_{x'})^{\mathsf{T}}\left(\mathbf{I}_d + 2\gamma(\Sigma_x + \Sigma_{x'})\right)^{-1}(\mu_x - \mu_{x'})\right).$$

Thus if $\Sigma_x = \mathrm{diag}(\Sigma_{x,1}, \ldots, \Sigma_{x,d})$ and $\Sigma_{x'} = \mathrm{diag}(\Sigma_{x',1}, \ldots, \Sigma_{x',d})$, then

$$\mathbb{E}_{Z_x \sim P_x, Z_{x'} \sim P_{x'}} \exp\left(-\gamma \|Z_x - Z_{x'}\|_2^2\right)$$
$$= \prod_{i=1}^{d} \left[\left(1 + 2\gamma(\Sigma_{x,i} + \Sigma_{x',i})\right)^{-1/2} \exp\left(-\gamma\left(1 + 2\gamma(\Sigma_{x,i} + \Sigma_{x',i})\right)^{-1}\left(\mu_{x,i} - \mu_{x',i}\right)^2\right)\right].$$

Hence we see that a Gaussian kernel
$$k_{\mathcal{Y}}(y, y') = \exp\left(-\gamma\|y - y'\|_2^2\right)$$
with inverse length scale $\gamma > 0$ on the space of targets $\mathcal{Y} = \mathbb{R}^d$ allows us to compute $\mathbb{E}_{Z_x \sim P_x} k_{\mathcal{Y}}(Z_x, y)$ and $\mathbb{E}_{Z_x \sim P_x, Z_{x'} \sim P_{x'}} k_{\mathcal{Y}}(Z_x, Z_{x'})$ analytically. Moreover, the Gaussian kernel is characteristic on $\mathbb{R}^d$ (Fukumizu et al., 2008). Hence, as discussed above, by choosing a kernel $k_{\mathcal{P}}$ that is non-zero almost surely we can guarantee that $\mathrm{KCE}_k = 0$ if and only if model $P$ is calibrated.

On the space of normal distributions, the 2-Wasserstein distance with respect to the Euclidean distance between $P_x = \mathcal{N}(\mu_x, \Sigma_x)$ and $P_{x'} = \mathcal{N}(\mu_{x'}, \Sigma_{x'})$ is given by

$$W_2^2(P_x, P_{x'}) = \|\mu_x - \mu_{x'}\|_2^2 + \mathrm{Tr}\left(\Sigma_x + \Sigma_{x'} - 2\left(\Sigma_{x'}^{1/2}\Sigma_x\Sigma_{x'}^{1/2}\right)^{1/2}\right),$$

which can be simplified to

$$W_2^2(P_x, P_{x'}) = \left\|\mu_x - \mu_{x'}\right\|_2^2 + \left\|\Sigma_x^{1/2} - \Sigma_{x'}^{1/2}\right\|_{\mathrm{Frob}}^2,$$

if $\Sigma_x\Sigma_{x'} = \Sigma_{x'}\Sigma_x$. This shows that the 2-Wasserstein distance is a Hilbertian metric on the space of normal distributions. Hence as discussed above, the choice
$$k_{\mathcal{P}}(P_x, P_{x'}) = \exp\left(-\lambda W_2^{\nu}(P_x, P_{x'})\right)$$
yields a valid kernel for all $\lambda > 0$ and $\nu \in (0, 2]$.

Thus for all $\lambda, \gamma > 0$ and $\nu \in (0, 2]$

$$k\big((p, y), (p', y')\big) = \exp\left(-\lambda W_2^{\nu}(p, p')\right) \exp\left(-\gamma\|y - y'\|_2^2\right)$$

is a valid kernel on the product space $\mathcal{P} \times \mathcal{Y}$ of normal distributions on $\mathbb{R}^d$ and $\mathbb{R}^d$ that allows to evaluate $h\big((p, y), (p', y')\big)$ analytically and guarantees that $\mathrm{KCE}_k = 0$ if and only if model $P$ is calibrated.

### D.2  LAPLACE DISTRIBUTIONS

Assume that $\mathcal{Y} = \mathbb{R}$ and $\mathcal{P} = \{\mathcal{L}(\mu, \beta) \colon \mu \in \mathbb{R}, \beta > 0\}$, i.e., the model outputs Laplace distributions $P_X = \mathcal{L}(\mu_X, \beta_X)$ with probability density function

$$p_X(y) = \frac{1}{2\beta_X} \exp\left(-\beta_X^{-1}|y - \mu_X|\right)$$

for $y \in \mathcal{Y} = \mathbb{R}$. The distribution of these outputs is defined by the distribution of their mean $\mu_X$ and scale parameter $\beta_X$.

Let $P_x = \mathcal{L}(\mu_x, \beta_x) \in \mathcal{P}$, $y \in \mathcal{Y} = \mathbb{R}$, and $\gamma > 0$. If $\beta_x \neq \gamma^{-1}$, we have

$$\mathbb{E}_{Z_x \sim P_x} \exp\left(-\gamma|Z_x - y|\right)$$
$$= \left(\beta_x^2\gamma^2 - 1\right)^{-1}\left(\beta_x\gamma \exp\left(-\beta_x^{-1}|\mu_x - y|\right) - \exp\left(-\gamma|\mu_x - y|\right)\right).$$

Additionally, if $\beta_x = \gamma^{-1}$, the dominated convergence theorem implies

$$\mathbb{E}_{Z_x \sim P_x} \exp\left(-\gamma|Z_x - y|\right)$$
$$= \lim_{\gamma \to \beta_x^{-1}} \left(\beta_x^2\gamma^2 - 1\right)^{-1}\left(\beta_x\gamma \exp\left(-\beta_x^{-1}|\mu_x - y|\right) - \exp\left(-\gamma|\mu_x - y|\right)\right)$$
$$= \frac{1}{2}\left(1 + \gamma|\mu_x - y|\right) \exp\left(-\gamma|\mu_x - y|\right).$$

Let $P_{x'} = \mathcal{L}(\mu_{x'}, \beta_{x'})$ be another Laplace distribution. If $\beta_x \neq \gamma^{-1}$, $\beta_{x'} \neq \gamma^{-1}$, and $\beta_x \neq \beta_{x'}$, we obtain

$$\mathbb{E}_{Z_x \sim P_x, Z_{x'} \sim P_{x'}} \exp\left(-\gamma|Z_x - Z_{x'}|\right) = \frac{\gamma\beta_x^3}{(\beta_x^2\gamma^2 - 1)(\beta_x^2 - \beta_{x'}^2)} \exp\left(-\beta_x^{-1}|\mu_x - \mu_{x'}|\right)$$
$$+ \frac{\gamma\beta_{x'}^3}{(\beta_{x'}^2\gamma^2 - 1)(\beta_{x'}^2 - \beta_x^2)} \exp\left(-\beta_{x'}^{-1}|\mu_x - \mu_{x'}|\right)$$
$$+ \frac{1}{(\beta_x^2\gamma^2 - 1)(\beta_{x'}^2\gamma^2 - 1)} \exp\left(-\gamma|\mu_x - \mu_{x'}|\right).$$

As above, all other possible cases can be deduced by applying the dominated convergence theorem. More concretely,

- if $\beta_x = \beta_{x'} = \gamma^{-1}$, then

$$\mathbb{E}_{Z_x \sim P_x, Z_{x'} \sim P_{x'}} \exp\big(-\gamma|Z_x - Z_{x'}|\big)$$
$$= \frac{1}{8}\Big(3 + 3\gamma|\mu_x - \mu_{x'}| + \gamma^2|\mu_x - \mu_{x'}|^2\Big) \exp\big(-\gamma|\mu_x - \mu_{x'}|\big),$$

- if $\beta_x = \beta_{x'}$ and $\beta_x \neq \gamma^{-1}$, then

$$\mathbb{E}_{Z_x \sim P_x, Z_{x'} \sim P_{x'}} \exp\big(-\gamma|Z_x - Z_{x'}|\big) = \frac{1}{(\beta_x^2\gamma^2 - 1)^2} \exp\big(-\gamma|\mu_x - \mu_{x'}|\big)$$
$$+ \left(\frac{\gamma(\beta_x + |\mu_x - \mu_{x'}|)}{2(\beta_x^2\gamma^2 - 1)} - \frac{\beta_x\gamma}{(\beta_x^2\gamma^2 - 1)^2}\right) \exp\big(-\beta_x^{-1}|\mu_x - \mu_{x'}|\big),$$

- if $\beta_x \neq \beta_{x'}$ and $\beta_x = \gamma^{-1}$, then

$$\mathbb{E}_{Z_x \sim P_x, Z_{x'} \sim P_{x'}} \exp\big(-\gamma|Z_x - Z_{x'}|\big) = \frac{{\beta_{x'}}^3\gamma^3}{({\beta_{x'}}^2\gamma^2 - 1)^2} \exp\big(-{\beta_{x'}}^{-1}|\mu_x - \mu_{x'}|\big)$$
$$- \left(\frac{1 + \gamma|\mu_x - \mu_{x'}|}{2({\beta_{x'}}^2\gamma^2 - 1)} + \frac{{\beta_{x'}}^2\gamma^2}{({\beta_{x'}}^2\gamma^2 - 1)^2}\right) \exp\big(-\gamma|\mu_x - \mu_{x'}|\big),$$

- and if $\beta_x \neq \beta_{x'}$ and $\beta_{x'} = \gamma^{-1}$, then

$$\mathbb{E}_{Z_x \sim P_x, Z_{x'} \sim P_{x'}} \exp\big(-\gamma|Z_x - Z_{x'}|\big) = \frac{\beta_x^3\gamma^3}{(\beta_x^2\gamma^2 - 1)^2} \exp\big(-\beta_x^{-1}|\mu_x - \mu_{x'}|\big)$$
$$- \left(\frac{1 + \gamma|\mu_x - \mu_{x'}|}{2(\beta_x^2\gamma^2 - 1)} + \frac{\beta_x^2\gamma^2}{(\beta_x^2\gamma^2 - 1)^2}\right) \exp\big(-\gamma|\mu_x - \mu_{x'}|\big).$$

The calculations above show that by choosing a Laplacian kernel

$$k_{\mathcal{Y}}(y, y') = \exp\big(-\gamma|y - y'|\big)$$

with inverse length scale $\gamma > 0$ on the space of targets $\mathcal{Y} = \mathbb{R}$, we can compute $\mathbb{E}_{Z_x \sim P_x} k_{\mathcal{Y}}(Z_x, y)$ and $\mathbb{E}_{Z_x \sim P_x, Z_{x'} \sim P_{x'}} k_{\mathcal{Y}}(Z_x, Z_{x'})$ analytically. Additionally, the Laplacian kernel is characteristic on $\mathbb{R}$ (Fukumizu et al., 2008).

Since the Laplace distribution is an elliptically contoured distribution, we know from Gelbrich (1990, Corollary 2) that the 2-Wasserstein distance with respect to the Euclidean distance between $P_x = \mathcal{L}(\mu_x, \beta_x)$ and $P_{x'} = \mathcal{L}(\mu_{x'}, \beta_{x'})$ can be computed in closed form and is given by

$$W_2^2\big(P_x, P_{x'}\big) = (\mu_x - \mu_{x'})^2 + 2(\beta_x - \beta_{x'})^2.$$

Thus we see that the 2-Wasserstein distance is also a Hilbertian metric on the space of Laplace distributions, and hence

$$k_{\mathcal{P}}\big(P_x, P_{x'}\big) = \exp\big(-\lambda W_2^\nu(P_x, P_{x'})\big)$$

is a valid kernel for $0 < \nu \leq 2$ and all $\lambda > 0$.

Therefore, as discussed above, for all $\lambda, \gamma > 0$ and $\nu \in (0, 2]$

$$k\big((p, y), (p', y')\big) = \exp\big(-\lambda W_2^\nu(p, p')\big) \exp\big(-\gamma|y - y'|\big)$$

is a valid kernel on the product space $\mathcal{P} \times \mathcal{Y}$ of Laplace distributions and $\mathbb{R}$ that allows to evaluate $h\big((p, y), (p', y')\big)$ analytically and guarantees that $\text{KCE}_k = 0$ if and only if model $P$ is calibrated.

Assume that the model predicts mixture distributions, possibly with different numbers of components. A special case of this setting are ensembles of models, in which each ensemble member predicts a component of the mixture model.

Let $p, p' \in \mathcal{P}$ with $p = \sum_i \pi_i p_i$ and $p' = \sum_j \pi'_j p'_j$, where $\pi, \pi'$ are histograms and $p_i, p'_j$ are the mixture components. For kernel $k_{\mathcal{Y}}$ and $y \in \mathcal{Y}$ we obtain

$$\mathbb{E}_{Z \sim p}\, k_{\mathcal{Y}}(Z, y) = \sum_i \pi_i\, \mathbb{E}_{W \sim p_i}\, k_{\mathcal{Y}}(Z, y)$$

and

$$\mathbb{E}_{Z \sim p, Z' \sim p'}\, k_{\mathcal{Y}}(Z, Z') = \sum_{i,j} \pi_i \pi'_j\, \mathbb{E}_{Z \sim p_i, Z' \sim p'_j}\, k_{\mathcal{Y}}(Z, Z').$$

Of course, for these derivations to be meaningful, we require that they do not depend on the choice of histograms $\pi, \pi'$ and mixture components $p_i, p'_j$.

**Definition D.1 (see Yakowitz & Spragins (1968)).** A family $\mathcal{P}$ of finite mixture models is called identifiable if two mixtures $p = \sum_{i=1}^{K} \pi_i p_i \in \mathcal{P}$ and $p' = \sum_{j=1}^{K'} \pi'_j p'_j \in \mathcal{P}$, written such that all $p_i$ and all $p'_j$ are pairwise distinct, are equal if and only if $K = K'$ and the indices can be reordered such that for all $k \in \{1, \dots, K\}$ there exists some $k' \in \{1, \dots, K\}$ with $\pi_k = \pi'_{k'}$ and $p_k = p'_{k'}$.

Clearly, if $\mathcal{P}$ is identifiable, then the derivations above do not depend on the choice of histograms and mixture components. Prominent examples of identifiable mixture models are Gaussian mixture models and mixture models of families of products of exponential distributions (Yakowitz & Spragins, 1968).

Moreover, similar to optimal transport for Gaussian mixture models by Chen et al. (2019; 2020); Delon & Desolneux (2020), we can consider metrics of the form

$$\inf_{w \in \Pi(\pi, \pi')} \left( \sum_{i,j} w_{i,j}\, c^s(p_i, p'_j) \right)^{1/s},$$

where

$$\Pi(\pi, \pi') = \left\{ w \colon \sum_i w_{i,j} = \pi'_j \wedge \sum_j w_{i,j} = \pi_i \wedge \forall i, j \colon w_{i,j} \geq 0 \right\}$$

are the couplings of $\pi$ and $\pi'$, and $c(\cdot, \cdot)$ is a cost function between the components of the mixture model.

**Theorem D.1.** *Let $\mathcal{P}$ be a family of finite mixture models that is identifiable in the sense of Definition D.1, and let $s \in [1, \infty)$.*

*If $d(\cdot, \cdot)$ is a (Hilbertian) metric on the space of mixture components, then the Mixture Wasserstein distance of order $s$ defined by*

$$\mathrm{MW}_s(p, p') := \inf_{w \in \Pi(\pi, \pi')} \left( \sum_{i,j} w_{i,j}\, d^s(p_i, p'_j) \right)^{1/s}, \tag{D.2}$$

*is a (Hilbertian) metric on $\mathcal{P}$.*

*Proof.* First of all, note that for all $p, p' \in \mathcal{P}$ an optimal coupling $\hat{w}$ exists (Villani, 2009, Theorem 4.1). Moreover, $\sum_{i,j} \hat{w}_{i,j}\, d^s(p_i, p'_j) \geq 0$, and hence $\mathrm{MW}_s(p, p')$ exists. Moreover, since $\mathcal{P}$ is identifiable, we see that $\mathrm{MW}_s(p, p')$ does not depend on the choice of histograms and mixture components. Thus $\mathrm{MW}_s$ is well-defined.

Clearly, for all $p, p' \in \mathcal{P}$ we have $\mathrm{MW}_s(p, p') \geq 0$ and $\mathrm{MW}_s(p, p') = \mathrm{MW}_s(p', p)$. Moreover,

$$\mathrm{MW}_s^s(p, p) = \min_{w \in \Pi(\pi, \pi)} \sum_{i,j} w_{i,j}\, d^s(p_i, p_j) \leq \sum_{i,j} \pi_i \delta_{i,j}\, d^s(p_i, p_j)$$

$$= \sum_i \pi_i\, d^s(p_i, p_i) = \sum_i \pi_i 0^2 = 0,$$

and hence $\mathrm{MW}_s(p, p) = 0$. On the other hand, let $p, p' \in \mathcal{P}$ with optimal coupling $\hat{w}$ with respect to $\pi$ and $\pi'$, and assume that $\mathrm{MW}_s(p, p') = 0$. We have

$$p = \sum_i \pi_i p_i = \sum_{i,j} \hat{w}_{i,j} p_i = \sum_{i,j \colon \hat{w}_{i,j} > 0} \hat{w}_{i,j} p_i.$$

Since $\mathrm{MW}_s(p, p') = 0$, we have $\hat{w}_{i,j} d^s(p_i, p'_j) = 0$ for all $i, j$, and hence $d^s(p_i, p'_j) = 0$ if $\hat{w}_{i,j} > 0$. Since $d$ is a metric, this implies $p_i = p'_j$ if $\hat{w}_{i,j} > 0$. Thus we get

$$p = \sum_{i,j \colon \hat{w}_{i,j} > 0} \hat{w}_{i,j} p_i = \sum_{i,j \colon \hat{w}_{i,j} > 0} \hat{w}_{i,j} p'_j = \sum_{i,j} \hat{w}_{i,j} p'_j = \sum_j \pi'_j p'_j = p'.$$

Function $\mathrm{MW}_s$ also satisfies the triangle inequality, following a similar argument as Chen et al. (2019). Let $p^{(1)}, p^{(2)}, p^{(3)} \in \mathcal{P}$ and denote the optimal coupling with respect to $\pi^{(1)}$ and $\pi^{(2)}$ by $\hat{w}^{(12)}$, and the optimal coupling with respect to $\pi^{(2)}$ and $\pi^{(3)}$ by $\hat{w}^{(23)}$. Define $w^{(13)}$ by

$$w_{i,k}^{(13)} := \sum_{j \colon \pi_j^{(2)} \neq 0} \frac{\hat{w}_{i,j}^{(12)} \hat{w}_{j,k}^{(23)}}{\pi_j^{(2)}}.$$

Clearly $w_{i,k}^{(13)} \geq 0$ for all $i, k$, and we see that

$$\sum_i w_{i,k}^{(13)} = \sum_i \sum_{j \colon \pi_j^{(2)} \neq 0} \frac{\hat{w}_{i,j}^{(12)} \hat{w}_{j,k}^{(23)}}{\pi_j^{(2)}} = \sum_{j \colon \pi_j^{(2)} \neq 0} \sum_i \frac{\hat{w}_{i,j}^{(12)} \hat{w}_{j,k}^{(23)}}{\pi_j^{(2)}}$$

$$= \sum_{j \colon \pi_j^{(2)} \neq 0} \frac{\pi_j^{(2)} \hat{w}_{j,k}^{(23)}}{\pi_j^{(2)}} = \sum_{j \colon \pi_j^{(2)} \neq 0} \hat{w}_{j,k}^{(23)} = \pi^{(3)} - \sum_{j \colon \pi_j^{(2)} = 0} \hat{w}_{j,k}^{(23)}$$

for all $k$. Since for all $j, k$, $\pi_j^{(2)} \geq \hat{w}_{j,k}^{(23)}$, we know that $\pi_j^{(2)} = 0$ implies $\hat{w}_{j,k}^{(23)} = 0$ for all $k$. Thus for all $k$

$$\sum_i w_{i,k}^{(13)} = \pi^{(3)}.$$

Similarly we obtain for all $i$

$$\sum_k w_{i,k}^{(13)} = \pi^{(1)}.$$

Thus $w^{(13)} \in \Pi(\pi^{(1)}, \pi^{(3)})$, and therefore by exploiting the triangle inequality for metric $d$ and the Minkowski inequality we get

$$\mathrm{MW}_s\big(p^{(1)}, p^{(3)}\big) \le \bigg( \sum_{i,k} w_{i,k}^{(13)} d^s\big(p_i^{(1)}, p_k^{(3)}\big) \bigg)^{1/s} = \bigg( \sum_{i,k} \sum_{j:\, \pi_j^{(2)} \ne 0} \frac{\hat{w}_{i,j}^{(12)} \hat{w}_{j,k}^{(23)}}{\pi_j^{(2)}} d^s\big(p_i^{(1)}, p_k^{(3)}\big) \bigg)^{1/s}$$

$$\le \bigg( \sum_{i,k} \sum_{j:\, \pi_j^{(2)} \ne 0} \frac{\hat{w}_{i,j}^{(12)} \hat{w}_{j,k}^{(23)}}{\pi_j^{(2)}} \big(d(p_i^{(1)}, p_j^{(2)}) + d(p_j^{(2)}, p_k^{(3)})\big)^s \bigg)^{1/s}$$

$$\le \bigg( \sum_{i,k} \sum_{j:\, \pi_j^{(2)} \ne 0} \frac{\hat{w}_{i,j}^{(12)} \hat{w}_{j,k}^{(23)}}{\pi_j^{(2)}} d^s\big(p_i^{(1)}, p_j^{(2)}\big) \bigg)^{1/s}$$

$$+ \bigg( \sum_{i,k} \sum_{j:\, \pi_j^{(2)} \ne 0} \frac{\hat{w}_{i,j}^{(12)} \hat{w}_{j,k}^{(23)}}{\pi_j^{(2)}} d^s\big(p_j^{(2)}, p_k^{(3)}\big) \bigg)^{1/s}$$

$$= \bigg( \sum_{i} \sum_{j:\, \pi_j^{(2)} \ne 0} \hat{w}_{i,j}^{(12)} d^s\big(p_i^{(1)}, p_j^{(2)}\big) \bigg)^{1/s}$$

$$+ \bigg( \sum_{k} \sum_{j:\, \pi_j^{(2)} \ne 0} \hat{w}_{i,k}^{(23)} d^s\big(p_j^{(2)}, p_k^{(3)}\big) \bigg)^{1/s}$$

$$\le \bigg( \sum_{i,j} \hat{w}_{i,j}^{(12)} d^s\big(p_i^{(1)}, p_j^{(2)}\big) \bigg)^{1/s} + \bigg( \sum_{j,k} \hat{w}_{i,k}^{(23)} d^s\big(p_j^{(2)}, p_k^{(3)}\big) \bigg)^{1/s}$$

$$= \mathrm{MW}_s\big(p^{(1)}, p^{(2)}\big) + \mathrm{MW}_s\big(p^{(2)}, p^{(3)}\big).$$

Thus $\mathrm{MW}_s$ is a metric, and it is just left to show that it is Hilbertian if $d$ is Hilbertian. Since $d$ is a Hilbertian metric, there exists a Hilbert space $\mathcal{H}$ and a mapping $\phi$ such that

$$d(x, y) = \|\phi(x) - \phi(y)\|_{\mathcal{H}}.$$

Let $r_1, \ldots, r_n \in \mathbb{R}$ with $\sum_i r_i = 0$ and $p^{(1)}, \ldots, p^{(n)} \in \mathcal{P}$. Denote the optimal coupling with respect to $\pi^{(i)}$ and $\pi^{(j)}$ by $\hat{w}^{(i,j)}$. Then we have

$$\sum_{i,j} r_i r_j \sum_{k,l} \hat{w}_{k,l}^{(i,j)} \|\phi(p_k^{(i)})\|_{\mathcal{H}}^2 = \sum_{i,k} r_i \|\phi(p_k^{(i)})\|_{\mathcal{H}}^2 \sum_{j} r_j \sum_{l} \hat{w}_{k,l}^{(i,j)}$$

$$= \sum_{i,k} r_i \|\phi(p_k^{(i)})\|_{\mathcal{H}}^2 \sum_{j} r_j \pi_k^{(i)} \tag{D.3}$$

$$= \sum_{i,k} r_i \pi_k^{(i)} \|\phi(p_k^{(i)})\|_{\mathcal{H}}^2 \sum_{j} r_j = 0,$$

and similarly

$$\sum_{i,j} r_i r_j \sum_{k,l} \hat{w}_{k,l}^{(i,j)} \|\phi(p_l^{(j)})\|_{\mathcal{H}}^2 = 0. \tag{D.4}$$

Moreover, for all $k, l$ we get

$$\sum_{i,j} r_i r_j \hat{w}_{k,l}^{(i,j)} \big\langle \phi\big(p_k^{(i)}\big), \phi\big(p_l^{(j)}\big) \big\rangle_{\mathcal{H}} = \Big\langle \sum_{i} r_i \sqrt{\hat{w}_{k,l}^{(i,j)}} \phi\big(p_k^{(i)}\big), \sum_{j} r_j \sqrt{\hat{w}_{k,l}^{(i,j)}} \phi\big(p_l^{(j)}\big) \Big\rangle_{\mathcal{H}}$$

$$= \Big\| \sum_{i} r_i \sqrt{\hat{w}_{k,l}^{(i,j)}} \phi\big(p_k^{(i)}\big) \Big\|_{\mathcal{H}}^2 \ge 0,$$

and hence

$$\sum_{i,j} r_i r_j \sum_{k,l} \hat{w}_{k,l}^{(i,j)} \big\langle \phi(p_k^{(i)}), \phi(p_l^{(j)}) \big\rangle_{\mathcal{H}} \ge 0, \tag{D.5}$$

and similarly

$$\sum_{i,j} r_i r_j \sum_{k,l} \hat{w}_{k,l}^{(i,j)} \left\langle \phi(p_l^{(j)}), \phi(p_k^{(i)}) \right\rangle_{\mathcal{H}} \geq 0. \tag{D.6}$$

Hence from Eqs. (D.3) to (D.6) we get

$$\begin{aligned}
\sum_{i,j} r_i r_j \mathrm{MW}_s^s(p^{(i)}, p^{(j)}) &= \sum_{i,j} r_i r_j \sum_{k,l} \hat{w}_{k,l}^{(i,j)} d^s \left( p_k^{(i)}, p_l^{(j)} \right) \\
&= \sum_{i,j} r_i r_j \sum_{k,l} \hat{w}_{k,l}^{(i,j)} \left\| \phi\left( p_k^{(i)} \right) - \phi\left( p_l^{(j)} \right) \right\|_{\mathcal{H}}^2 \\
&= \sum_{i,j} r_i r_j \sum_{k,l} \hat{w}_{k,l}^{(i,j)} \left\| \phi\left( p_k^{(i)} \right) \right\|_{\mathcal{H}}^2 \\
&\quad - \sum_{i,j} r_i r_j \sum_{k,l} \hat{w}_{k,l}^{(i,j)} \left\langle \phi\left( p_k^{(i)} \right), \phi\left( p_l^{(j)} \right) \right\rangle_{\mathcal{H}} \\
&\quad - \sum_{i,j} r_i r_j \sum_{k,l} \hat{w}_{k,l}^{(i,j)} \left\langle \phi\left( p_l^{(j)} \right), \phi\left( p_k^{(i)} \right) \right\rangle_{\mathcal{H}} \\
&\quad + \sum_{i,j} r_i r_j \sum_{k,l} \hat{w}_{k,l}^{(i,j)} \left\| \phi\left( p_l^{(j)} \right) \right\|_{\mathcal{H}}^2 \\
&\leq 0,
\end{aligned}$$

which shows that $\mathrm{MW}_s^s$ is a negative definite kernel (Berg et al., 1984, Definition 3.1.1). Since $0 < 1/s < \infty$, $\mathrm{MW}_s$ is a negative definite kernel as well (Berg et al., 1984, Corollary 3.2.10), which implies that metric $\mathrm{MW}_s$ is Hilbertian (Berg et al., 1984, Proposition 3.3.2). $\qquad\square$

Hence we can lift a Hilbertian metric for the mixture components to a Hilbertian metric for the mixture models. For instance, if the mixture components are normal distributions, then the 2-Wasserstein distance with respect to the Euclidean distance is a Hilbertian metric for the mixture components. When we lift it to the space $\mathcal{P}$ of Gaussian mixture models we obtain the $\mathrm{MW}_2$ metric proposed by Chen et al. (2019; 2020); Delon & Desolneux (2020). As shown by Delon & Desolneux (2020), the discrete formulation of $\mathrm{MW}_2$ obtained by our construction is equivalent to the definition

$$\mathrm{MW}_2^2(p, p') := \inf_{\gamma \in \Pi(p,p') \cap \mathrm{GMM}_{2n}(\infty)} \int_{\mathbb{R}^n \times \mathbb{R}^n} d^2(y, y') \, \mathrm{d}\gamma(y, y') \tag{D.7}$$

for two Gaussian mixtures $p, p'$ on $\mathbb{R}^n$, where $\Pi(p, p')$ are the couplings of $p$ and $p'$ (not of the histograms!) and $\mathrm{GMM}_{2n}(\infty) = \cup_{k \geq 0} \mathrm{GMM}_{2n}(k)$ is the set of all finite Gaussian mixture distributions on $\mathbb{R}^{2n}$. The construction of the discrete formulation as a solution to a constrained optimization problem similar to Eq. (D.7) can be generalized to mixtures of $t$-distributions. However, it is not possible for arbitrary mixture models such as mixtures of generalized Gaussian distributions, even though they are elliptically contoured distributions (Deledalle et al., 2018; Delon & Desolneux, 2020).

The optimal coupling of the discrete histograms can be computed efficiently using techniques from linear programming and optimal transport theory such as the network simplex algorithm and the Sinkhorn algorithm. As discussed above, if metric $d_{\mathcal{P}}$ is of the form in Eq. (D.2), functions of the form

$$k_{\mathcal{P}}(p, p') = \exp\left( -\lambda d_{\mathcal{P}}^\nu(p, p') \right)$$

are valid kernels on $\mathcal{P}$ for all $\lambda > 0$ and $\nu \in (0, 2]$.

Thus taken together, if $k_{\mathcal{Y}}$ is a characteristic kernel on the target space $\mathcal{Y}$ and $d(\cdot, \cdot)$ is a Hilbertian metric on the space of mixture components, then for all $s \in [1, \infty)$, $\lambda > 0$, and $\nu \in (0, 2]$

$$k\big((p, y), (p', y')\big) = \exp\left( -\lambda \mathrm{MW}_s^\nu(p, p') \right) k_{\mathcal{Y}}(y, y')$$

is a valid kernel on the product space $\mathcal{P} \times \mathcal{Y}$ of mixture distributions and targets that allows to evaluate $h\big((p, y), (p', y')\big)$ analytically and guarantees that $\mathrm{KCE}_k = 0$ if and only if model $P$ is calibrated.

# E CLASSIFICATION AS A SPECIAL CASE

We show that the calibration error introduced in Definition 2 is a generalization of the calibration error for classification proposed by Widmann et al. (2019). Their formulation of the calibration error is based on a weighted sum of class-wise discrepancies between the left hand side and right hand side of Definition 1, where the weights are output by a vector-valued function of the predictions. Hence their framework can only be applied to finite target spaces, i.e., if $|\mathcal{Y}| < \infty$.

Without loss of generality, we assume that $\mathcal{Y} = \{1, \ldots, d\}$ for some $d \in \mathbb{N} \setminus \{1\}$. In our notation, the previously defined calibration error, denoted by CCE (classification calibration error), with respect to a function space $\mathcal{G} \subset \{f \colon \mathcal{P} \to \mathbb{R}^d\}$ is given by

$$\mathrm{CCE}_{\mathcal{G}} \coloneqq \sup_{g \in \mathcal{G}} \left| \mathbb{E}_{P_X} \left( \sum_{y \in \mathcal{Y}} \big( \mathbb{P}(Y = y | P_X) - P_X(\{y\}) \big) g_y(P_X) \right) \right|.$$

For the function class

$$\mathcal{F} \coloneqq \big\{ f \colon \mathcal{P} \times \mathcal{Y} \to \mathbb{R}, (p, y) \mapsto g_y(p) \big| g \in \mathcal{G} \big\}$$

we get

$$\mathrm{CCE}_{\mathcal{G}} = \sup_{f \in \mathcal{F}} \big| \mathbb{E}_{P_X, Y} f(P_X, Y) - \mathbb{E}_{P_X, Z_X} f(P_X, Z_X) \big| = \mathrm{CE}_{\mathcal{F}}.$$

Similarly, for every function class $\mathcal{F} \subset \{f \colon \mathcal{P} \times \mathcal{Y} \to \mathbb{R}\}$, we can define the space

$$\mathcal{G} \coloneqq \Big\{ g \colon \mathcal{P} \to \mathbb{R}^d, p \mapsto \big( f(p, 1), \ldots, f(p, d) \big)^{\mathsf{T}} \Big| f \in \mathcal{F} \Big\},$$

for which

$$\mathrm{CE}_{\mathcal{F}} = \sup_{g \in \mathcal{G}} \left| \mathbb{E}_{P_X} \left( \sum_{y \in \mathcal{Y}} \big( \mathbb{P}(Y = y | P_X) - P_X(\{y\}) \big) g_y(P_X) \right) \right| = \mathrm{CCE}_{\mathcal{G}}.$$

Thus both definitions are equivalent for classification models but the structure of the employed function classes differs. The definition of CCE is based on vector-valued functions on the probability simplex whereas the formulation presented in this paper uses real-valued function on the product space of the probability simplex and the targets.

An interesting theoretical aspect of this difference is that in the case of KCE we consider real-valued kernels on $\mathcal{P} \times \mathcal{Y}$ instead of matrix-valued kernels on $\mathcal{P}$, as shown by the following comparison. By $e_i \in \mathbb{R}^d$ we denote the $i$th unit vector, and for a prediction $p \in \mathcal{P}$ its representation $v_p \in \mathbb{R}^d$ in the probability simplex is defined as

$$(v_p)_y = p(\{y\})$$

for all targets $y \in \mathcal{Y}$.

Let $k \colon (\mathcal{P} \times \mathcal{Y}) \times (\mathcal{P} \times \mathcal{Y}) \to \mathbb{R}$. We define the matrix-valued function $K \colon \mathcal{P} \times \mathcal{P} \to \mathbb{R}^{d \times d}$ by

$$\big[ K(p, p') \big]_{y, y'} = k\big( (p, y), (p', y') \big)$$

for all $y, y' \in \mathcal{Y}$ and $p, p' \in \mathcal{P}$. From the positive definiteness of kernel $k$ it follows that $K$ is a matrix-valued kernel (Micchelli & Pontil, 2005, Definition 2). We obtain

$$\begin{aligned}
\mathrm{SKCE}_k &= \mathbb{E}_{P_X, Y, P_{X'}, Y'} \big[ K(P_X, P_{X'}) \big]_{Y, Y'} - 2 \mathbb{E}_{P_X, Y, P_{X'}, Z_{X'}} \big[ K(P_X, P_{X'}) \big]_{Y, Z_{X'}} \\
&\quad + \mathbb{E}_{P_X, Z_X, P_{X'}, Z_{X'}} \big[ K(P_X, P_{X'}) \big]_{Z_X, Z_{X'}} \\
&= \mathbb{E}_{P_X, Y, P_{X'}, Y'} \, e_Y^{\mathsf{T}} K(P_X, P_{X'}) e_{Y'} - 2 \mathbb{E}_{P_X, Y, P_{X'}, Y'} \, e_Y^{\mathsf{T}} K(P_X, P_{X'}) v_{P_{X'}} \\
&\quad + \mathbb{E}_{P_X, Y, P_{X'}, Y'} \, v_{P_X}^{\mathsf{T}} K(P_X, P_{X'}) v_{P_{X'}} \\
&= \mathbb{E}_{P_X, Y, P_{X'}, Y'} \, (e_Y - v_{P_X})^{\mathsf{T}} K(P_X, P_{X'}) (e_{Y'} - v_{P_{X'}}),
\end{aligned}$$

which is exactly the result by Widmann et al. (2019) for matrix-valued kernels.

As a concrete example, Widmann et al. (2019) used a matrix-valued kernel of the form $(p, p') \mapsto \exp(-\gamma \|p - p'\|) \mathbf{I}_d$ in their experiments. In our formulation this corresponds to the real-valued tensor product kernel $\big( (p, y), (p', y') \big) \mapsto \exp(-\gamma \|p - p'\|) \delta_{y, y'}$.

## F  TEMPERATURE SCALING

Since many modern neural network models for classification have been demonstrated to be uncalibrated (Guo et al., 2017), it is of high practical interest being able to improve calibration of predictive models. Generally, one distinguishes between calibration techniques that are applied during training and post-hoc calibration methods that try to calibrate an existing model after training.

Temperature scaling (Guo et al., 2017) is a simple calibration method for classification models with only one scalar parameter. Due to its simplicity it can trade off calibration of different classes (Kull et al., 2019), but conveniently it does not change the most-confident prediction and hence does not affect the accuracy of classification models with respect to the 0-1 loss.

In regression, common post-hoc calibration methods are based on quantile binning and hence insufficient for our framework. Song et al. (2019) proposed a calibration method for regression models with real-valued targets, based on a special case of Definition 1. This calibration method was shown to perform well empirically but is computationally expensive and requires users to choose hyperparameters for a Gaussian process model and its variational inference. As a simpler alternative, we generalize temperature scaling to arbitrary predictive models in the following way.

**Definition F.1.** Let $P_x$ be the output of a probabilistic predictive model $P$ for feature $x$. If $P_x$ has probability density function $p_x$ with respect to a reference measure $\mu$, then temperature scaling with respect to $\mu$ with temperature $T > 0$ yields a new output $Q_x$ whose probability density function $q_x$ with respect to $\mu$ satisfies

$$q_x \propto p_x^{1/T}.$$

The notion for classification models given by Guo et al. (2017) can be recovered by choosing the counting measure on the classes as reference measure.

For some exponential families on $\mathbb{R}^d$ we obtain particularly simple transformations with respect to the Lebesgue measure $\lambda^d$ that keep the type of predicted distribution and its mean invariant. Hence in contrast to other calibration methods, for these models temperature scaling yields analytically tractable distributions and does not negatively impact the accuracy of the models with respect to the mean squared error and the mean absolute error.

For instance, temperature scaling of multivariate power exponential distributions (Gómez et al., 1998) in $\mathbb{R}^d$, of which multivariate normal distributions are a special case, with respect to $\lambda^d$ corresponds to multiplication of their scale parameter with $T^{1/\beta}$, where $\beta$ is the so-called kurtosis parameter (Gómez-Sánchez-Manzano et al., 2008). For normal distributions, this corresponds to multiplication of the covariance matrix with $T$.

Similarly, temperature scaling of Beta and Dirichlet distributions with respect to reference measure

$$\mu(\mathrm{d}x) := x^{-1}(1 - x)^{-1} \mathbb{1}_{(0,1)}(x) \lambda^1(\mathrm{d}x)$$

and

$$\mu(\mathrm{d}x) := \left( \prod_{i=1}^{d} x_i^{-1} \right) \mathbb{1}_{(0,1)^d}(x) \lambda^d(\mathrm{d}x),$$

respectively, corresponds to division of the canonical parameters of these distributions by $T$ without affecting the predicted mean value.

All in all, we see that temperature scaling for general predictive models preserves some of the nice properties for classification models. For some exponential families such as normal distributions reference measure $\mu$ can be chosen such that temperature scaling is a simple transformation of the parameters of the predicted distributions (and hence leaves the considered model class invariant) that does not affect accuracy of these models with respect to the mean squared error and the mean absolute error.

## G  EXPECTED CALIBRATION ERROR FOR COUNTABLY INFINITE DISCRETE TARGET SPACES

In literature, $\mathrm{ECE}_d$ and $\mathrm{MCE}_d$ are defined for binary and multi-class classification problems (Guo et al., 2017; Naeini et al., 2015; Vaicenavicius et al., 2019). For common distance measures on the

probability simplex such as the total variation distance and the squared Euclidean distance, $\mathrm{ECE}_d$ and $\mathrm{MCE}_d$ can be formulated as a calibration error in the framework of Widmann et al. (2019), which is a special case of the framework proposed in this paper for binary and multi-class classification problems.

In contrast to previous approaches, our framework handles countably infinite discrete target spaces as well. For every problem with countably infinitely many targets, such as, e.g., Poisson regression, there exists an equivalent regression problem on the set of natural numbers. Hence without loss of generality we assume $\mathcal{Y} = \mathbb{N}$. Denote the space of probability distributions on $\mathbb{N}$, the infinite dimensional probability simplex, with $\Delta^\infty$. Clearly, $\Delta^\infty$ can be viewed as a subspace of the sequence space $\ell^1$ that consists of all sequences $x = (x_n)_{n \in \mathbb{N}}$ with $x_n \geq 0$ for all $n \in \mathbb{N}$ and $\|x\|_1 = 1$.

**Theorem G.1.** *Let $1 < p < \infty$ with Hölder conjugate $q$. If*

$$\mathcal{F} := \{f \colon \Delta^\infty \times \mathbb{N} \to \mathbb{R} \mid \mathbb{E}_{P_X} \|(f(P_X, n))_{n \in \mathbb{N}}\|_p^p \leq 1\},$$

*then*

$$\mathrm{CE}_{\mathcal{F}}^q = \mathbb{E}_{P_X} \|\mathbb{P}(Y|P_X) - P_X\|_q^q.$$

*Let $\mu$ be the law of $P_X$. If $\mathcal{F} := \{f \colon \Delta^\infty \times \mathbb{N} \to \mathbb{R} \mid \mathbb{E}_{P_X} \|(f(P_X, n))_{n \in \mathbb{N}}\|_1 \leq 1\}$, then*

$$\mathrm{CE}_{\mathcal{F}} = \mu\text{-}\operatorname*{ess\,sup}_{\xi \in \Delta^\infty} \sup_{y \in \mathbb{N}} |\mathbb{P}(Y = y|P_X = \xi) - \xi(\{y\})|.$$

*Moreover, if $\mathcal{F} = \{f \colon \Delta^\infty \times \mathbb{N} \to \mathbb{R} \mid \mu\text{-}\operatorname{ess\,sup}_{\xi \in \Delta^\infty} \sup_{y \in \mathbb{N}} |f(\xi, y)| \leq 1\}$, then*

$$\mathrm{CE}_{\mathcal{F}} = \mathbb{E}_{P_X} \|\mathbb{P}(Y|P_X) - P_X\|_1.$$

*Proof.* Let $1 \leq p \leq \infty$, and let $\mu$ be the law of $P_X$ and $\nu$ be the counting measure on $\mathbb{N}$. Since both $\mu$ and $\nu$ are $\sigma$-finite measures, the product measure $\mu \otimes \nu$ is uniquely determined and $\sigma$-finite as well. Using these definitions, we can reformulate $\mathcal{F}$ as

$$\mathcal{F} = \{f \in L^p(\Delta^\infty \times \mathbb{N}; \mu \otimes \nu) \mid \|f\|_{p;\mu \otimes \nu} \leq 1\}.$$

Define the function $\delta \colon \Delta^\infty \times \mathbb{N} \to \mathbb{R}$ $(\mu \otimes \nu)$-almost surely by

$$\delta(\xi, y) := \mathbb{P}(Y = y \mid P_X = \xi) - \xi(\{y\}).$$

Note that $\delta$ is well-defined since we assume that all singletons on $\Delta^\infty$ are $\mu$-measurable. Moreover, $\delta \in L^q(\Delta^\infty \times \mathbb{N}; \mu \otimes \nu)$, which follows from $(\xi, y) \mapsto \mathbb{P}(Y = y \mid P_X = \xi)$ and $(\xi, y) \mapsto \xi(\{y\})$ being functions in $L^q(\Delta^\infty \times \mathbb{N}; \mu \otimes \nu)$.

Since $\mu \otimes \nu$ is a $\sigma$-finite measure, the extremal equality of Hölder's inequality implies that

$$\mathrm{CE}_{\mathcal{F}} = \sup_{f \in \mathcal{F}} \mathbb{E}_{P_X, Y} f(P_X, Y) - \mathbb{E}_{P_X, Z_X} f(P_X, Z_X)$$

$$= \sup_{f \in \mathcal{F}} \left| \mathbb{E}_{P_X, Y} f(P_X, Y) - \mathbb{E}_{P_X, Z_X} f(P_X, Z_X) \right|$$

$$= \sup_{f \in \mathcal{F}} \left| \int_{\Delta^\infty \times \mathbb{N}} f(\xi, y) \delta(\xi, y) \, (\mu \otimes \nu)(\mathrm{d}(\xi, y)) \right|$$

$$= \|\delta\|_{q;\mu \otimes \nu}.$$

Note that the second equality follows from the symmetry of the function spaces $\mathcal{F}$: for every $f \in \mathcal{F}$, also $-f \in \mathcal{F}$.

Hence for $1 < p \leq \infty$, we obtain

$$\mathrm{CE}_{\mathcal{F}}^q = \int_{\Delta^\infty \times \mathbb{N}} |\delta(\xi, y)|^q \, (\mu \otimes \nu)(\mathrm{d}(\xi, y))$$

$$= \mathbb{E}_{P_X} \|(\delta(P_X, y))_{y \in \mathbb{N}}\|_q^q = \mathbb{E}_{P_X} \|\mathbb{P}(Y|P_X) - P_X\|_q^q.$$

For $p = 1$, we get

$$\mathrm{CE}_{\mathcal{F}} = \mu\text{-}\operatorname*{ess\,sup}_{\xi \in \Delta^\infty} \sup_{y \in \mathbb{N}} |\delta(\xi, y)| = \mu\text{-}\operatorname*{ess\,sup}_{\xi \in \Delta^\infty} \sup_{y \in \mathbb{N}} |\mathbb{P}(Y = y|P_X = \xi) - \xi(\{y\})|,$$

which concludes the proof. $\qquad\square$

We see that our framework deals with countably infinite discrete target spaces seamlessly whereas the previously proposed framework by Widmann et al. (2019) is not applicable to such spaces. It is mathematically pleasing to see that for countably infinite discrete targets the calibration errors obtained in Theorem G.1 within our framework coincide with the natural generalization of $\mathrm{ECE}_d$ and $\mathrm{MCE}_d$ given in Appendix B.2.

