# OpenReview forum: "Calibration tests beyond classification"
_ICLR.cc/2021/Conference — ICLR 2021 Poster_

### Official Review · AnonReviewer4 · 2020-10-26
**Experiments not convincing enough**

**Rating:** 5
**Confidence:** 3

**Review:**

The authors define (Definition 2) a generalized form of calibration error for any model with probabilistic output and any output space (binary classification, multiclass classification, real-valued regression, ordinal regression, structured prediction, etc.).

The main novelty of Definition 2 seems to be the introduction of a dummy variable $Z_X$ which is sampled from the predicted distribution $P_X$ over the target space, in order to compute an integral probability metric that compares the joint distribution of $(P_X, Y)$ to the joint distribution of $(P_X, Z_X)$. The authors note that in some cases, $Z_X$ can be integrated out of the definition either analytically or using numerical integration. It is not clear, from my limited background knowledge in the related work, why it should be helpful to introduce a dummy variable and then integrate it out.

At first reading, Definition 2 seems too general to be useful in practice, as it requires the choice of a space of functions $\mathcal{F}$. The authors argue (with details in the appendices, which are not provided to me at this time) that it generalizes several previous definitions of calibration error including the maximum mean discrepancy, the total variation distance, the Kantorovich distance, and the Dudley metric.

Section 3 went beyond my area of expertise and beyond my comprehension; I feel unqualified to provide an informed review of this section. I think some reference to kernels or RKHSs should be made in the title or the abstract.

The experiment in Section 5.2 demonstrates the utility of the proposed SKCE calibration metric, alongside more common metrics like negative log-likelihood (NLL) and mean squared error (MSE). It appears that the SKCE curves (for both training and test data) have a very similar shape to the NLL curves, so it's not clear what benefit SKCE provides above and beyond the more common and easily-computed NLL. I would have liked to see more convincing experimental evidence of the marginal benefit of this approach beyond common calibration metrics.

Regarding the significance of the work, I can add that in practice, I find that relatively few ML users are concerned with the calibration of their models, and these are entirely restricted to problems of classification (almost always binary classification) or quantile regression. The novelty of this work seems to lie mostly in its applicability beyond these types problems: the authors write, "A key contribution of this paper is a new framework that is applicable to multivariate regression, as well as... discrete ordinal or more complex (e.g., graph-structured) [output]," so I venture a guess that the intended audience for this work is relatively small.

**Minor comments**

In equation (2), I believe the RHS should be $\max P_X$ instead of $\arg \max P_X$.

Section 2, paragraph 2, you wrote "instead of the discrepancy between the conditional distributions $\mathbb{P}(Y | X)$ and $P_X$." Did you mean to write $\mathbb{P}(Y | P_X)$ instead of $\mathbb{P}(Y | X)$? That would make more sense to me, since the sentence would refer to comparing the LHS and RHS of Equation (1).

---

> ### Comment · Area_Chair1 · 2020-11-12
> **users and calibration of models**
>
> So, use of this approach does raise questions, as you say.  I'd guess that regression folks don't use calibration much because they have a hard time at it.  So this paper, possibly, fills a need.  And calibration definitely has critical uses for managing uncertainty and in doing active learning.

---

> ### Author Response · Authors · 2020-11-17
> **Author comments (I/II)**
>
> We thank you for your comments.
>
> *The main novelty of Definition 2 seems to be the introduction of a dummy variable $Z_X$ which is sampled from the predicted distribution $P_X$ over the target space, in order to compute an integral probability metric that compares the joint distribution of $(P_X, Y)$ to the joint distribution of $(P_X, Z_X)$. The authors note that in some cases, $Z_X$ can be integrated out of the definition either analytically or using numerical integration. It is not clear, from my limited background knowledge in the related work, why it should be helpful to introduce a dummy variable and then integrate it out.*
>
> Conceptually, the main advantage of introducing the artificial random variable is that it allows us to recast the calibration condition in the language of two-sample tests by comparing two distributions. With this reformulation we avoid checking the almost sure equality of the predicted distributions and corresponding conditional distributions of targets explicitly, which is challenging in particular for general probabilistic predictive models. By doing so we can exploit existing results from the MMD literature, however in an unusual and special setting since the conditional distribution of the artificial random variable is known (typically we would only be given samples from the two distributions). In the MMD case the special setup can be exploited in the estimators and tests by integrating out $Z_X$, but we nevertheless work with the same probabilistic (re-)formulation of the calibration error.
>
> *The experiment in Section 5.2 demonstrates the utility of the proposed SKCE calibration metric, alongside more common metrics like negative log-likelihood (NLL) and mean squared error (MSE). It appears that the SKCE curves (for both training and test data) have a very similar shape to the NLL curves, so it's not clear what benefit SKCE provides above and beyond the more common and easily-computed NLL. I would have liked to see more convincing experimental evidence of the marginal benefit of this approach beyond common calibration metrics.*
>
> We want to emphasize that neither log-likelihood nor mean squared error are calibration metrics. As shortly mentioned in the discussion, the decomposition of these scoring rules as sum of a so-called resolution term (which quantifies the sharpness of the predictions), a reliability term (which is a specific instance of the expected calibration error and hence quantifies calibration), and an entropy term (which quantifies the inherent uncertainty of the targets) shows that these metrics are evaluation metrics of probabilistic predictive models but not calibration metrics: models can trade off calibration for sharpness, i.e., a less calibrated but sharper model might yield a smaller NLL or MSE than a calibrated but less sharp model. One main advantage of our framework is that it provides a principled way for quantifying ONLY calibration for any probabilistic predictive model which previously was only possible for specific models such as classification models.
>
> Thus in our opinion the SKCE provides an additional benefit and serves a different purpose than common evaluation metrics such as NLL and MSE. Moreover, while the shape of the NLL, MSE, and SKCE is similar (which can be seen as promising as well since it indicates that the SKCE does not behave in completely unexpected and uncommon ways), the plots show that the models are ranked differently by the different metrics. For instance, the models with the smallest SKCE can be found at an earlier iteration than the best models according to NLL or MSE. This indicates that the model overfits in terms of calibration, before we can see any clear indication of overfitting in terms of NLL. Monitoring the SKCE in addition to the NLL can thus provide additional valuable information regarding the model during the training procedure.

---

> ### Author Response · Authors · 2020-11-17
> **Author comments (II/II)**
>
> *Regarding the significance of the work, I can add that in practice, I find that relatively few ML users are concerned with the calibration of their models, and these are entirely restricted to problems of classification (almost always binary classification) or quantile regression. The novelty of this work seems to lie mostly in its applicability beyond these types problems: the authors write, "A key contribution of this paper is a new framework that is applicable to multivariate regression, as well as... discrete ordinal or more complex (e.g., graph-structured) [output]," so I venture a guess that the intended audience for this work is relatively small.*
>
> We agree with this observation. However, to the best of our knowledge there are no calibration evaluation techniques available that go beyond classification problems and quantile regression. This makes it difficult for practitioners to reason about and evaluate calibration, even if they do care about it. We are convinced that calibration evaluation will be adopted and performed also in more general and complex problem settings if tools such as the methods proposed in this paper become available to practitioners.
>
> *In equation (2), I believe the RHS should be $\max P_X$ instead of $\arg \max P_X$.*
>
> *Section 2, paragraph 2, you wrote "instead of the discrepancy between the conditional distributions $\mathbb{P}(Y | X)$ and $P_X$." Did you mean to write $\mathbb{P}(Y | P_X)$ instead of $\mathbb{P}(Y | X)$? That would make more sense to me, since the sentence would refer to comparing the LHS and RHS of Equation (1).*
>
> Thank you for spotting these errors! Indeed, your assumptions are correct. We fixed these mistakes in the latest revision of our paper.

---

### Official Review · AnonReviewer1 · 2020-10-27
**A very nice piece of work**

**Rating:** 9
**Confidence:** 4

**Review:**

This paper addresses probabilistic data driven model calibration, i.e. aligning predicted target probabilities with actual ones.

This problem has been extensively studied for classification tasks but solutions for regression tasks have limitations as illustrated by Fig.1. The authors intend to fill this gap and introduce a general kernel-based calibration framework that subsumes other ones previously defined for classification. The contribution of the authors is thus clearly stated and positioned w.r.t. prior arts.

The authors start by proposing an alternative definition of calibration (Def.2) in order to cast the problem into integral probability metrics. It is this re-definition of the problem that allow them to encompass prior arts as special cases.
For a number of practical and theoretical reasons, the authors focus on a special case of this framework which involves the computation of the MMD as a metric.

Based on the MMD literature but also relying on the structure of their problem (where the auxiliary variable can be marginalized out), the authors provide several consistent estimator with known rates in dataset size.

The validity of the proposed estimates is assessed through convincing numerical experiments that involve a calibration test.

I honestly do not have much critic to address to this work which seems to have reached a level of maturity perfectly adapted for publication in ICLR. The only damper is that the proposed methodology allows to detect miscalibration not yet to cure it. However, the authors seem to have some ideas on that too as mentioned in their conclusion.

---

> ### Author Response · Authors · 2020-11-17
> **Author comments**
>
> We thank you for your comments and are very happy to hear that you found the paper interesting and well written.
>
> *I honestly do not have much critic to address to this work which seems to have reached a level of maturity perfectly adapted for publication in ICLR. The only damper is that the proposed methodology allows to detect miscalibration not yet to cure it. However, the authors seem to have some ideas on that too as mentioned in their conclusion.*
>
> In our opinion, it is mandatory to be able to detect miscalibration before curing it, and therefore in this paper we focused on evaluations and tests of calibration within the proposed framework. However, we agree that follow-up questions such as how to obtain calibrated models, possibly using the kernel estimators, is indeed an interesting topic for future work. As discussed in the conclusion, the differentiability of the kernel estimators might allow to incorporate the framework into the training procedure which would also demonstrate its usefulness for practical applications more clearly.

---

### Official Review · AnonReviewer2 · 2020-11-01
**Well-written and an elegant idea**

**Rating:** 7
**Confidence:** 4

**Review:**

Summary:
The authors present an approach for testing calibration in conditional probability estimation models. They build on a line of work in the kernel estimation literature assessing whether the conditional distributions are well calibrated (i.e. P(Y | f(X)) = f(X), where f is some predictive model). They develop an MMD kernel estimator and expand on practical choices of kernels that are computationally tractable. They then derive an asymptotic null distribution for calibrated models, enabling control over the error rate when labeling a model uncalibrated. A few simulation studies are done with neural networks to show the applicability of the method.

Review:
This is an excellently written paper. The intro and first few chapters are a joy to read and really explain the problem well. There is a lot of nuance to calibration, so I really appreciated the precision and clarity in the exposition.

The idea itself also seems quite elegant. Generalizing a previously published kernel approach from only discrete distributions to handle a more general class of problems may seem like a small conceptual step. However, I think the authors did a good job explaining the challenges of this extension. The resulting estimators are now applicable to many more problems than the existing work.

Note I am not an expert in kernel learning, so I have not evaluated the proofs for correctness.

My main issue comes with the lack of empirical studies. The toy problem is not terribly interesting and does not reveal any particular insight. It leads me to believe that maybe this is not that useful of a method, since the authors did not have anywhere that they could apply it to and derive meaningful insights or uses. The comment in the conclusion about the differentiability of their kernels is interesting and I think incorporating this into the training procedure could potentially show some very clear pragmatic use of this method.

Overall, I like the paper. It is clearly written and presents what I think is an interesting and novel idea.

---

> ### Comment · Area_Chair1 · 2020-11-12
> **on the empirical studies**
>
> Must admit, I did notice the experimental work was on rather restricted data.  This tends to happen with theoretical work, but I agree, I'd be much happier to see how this works on larger data.  The beauty of this approach is opening up calibration to a broader class of models, but we have to show it works.

---

> ### Author Response · Authors · 2020-11-17
> **Author comments**
>
> We thank you for your comments and are very happy to hear that you found the paper interesting and well written.
>
> *My main issue comes with the lack of empirical studies. The toy problem is not terribly interesting and does not reveal any particular insight. It leads me to believe that maybe this is not that useful of a method, since the authors did not have anywhere that they could apply it to and derive meaningful insights or uses.*
>
> In our empirical studies we wanted to focus on two points:
> 1. an experimental confirmation of the derived theoretical properties of the kernel-based estimators and hypothesis tests
> 2. a demonstration of how the framework can be applied to neural network models and ensembles of neural network models
>
> Synthetic models are required to empirically validate the theoretically expected statistical
> properties and computational efficiency of the estimators and tests (section 5.1), since the experiments have to be performed with (un)calibrated models.
>
> For demonstrating the application of our framework we deliberately chose a well-known regression problem from the statistics literature. Moreover, we tried to highlight that the framework can be applied to ensemble models as well.
>
> *The comment in the conclusion about the differentiability of their kernels is interesting and I think incorporating this into the training procedure could potentially show some very clear pragmatic use of this method.*
>
> We agree that incorporating the framework into the training procedure might demonstrate its usefulness for practical purposes more clearly. However, in our opinion, it is mandatory to be able to detect miscalibration before curing it, and therefore in this paper we focused on evaluations and tests of calibration within the proposed framework. Follow-up questions such as how to obtain calibrated models, possibly using the kernel estimators, is indeed an interesting topic for future work.

---

### Decision · Program_Chairs · 2021-01-07
**Final Decision**

**Decision:**

Accept (Poster)

**Comment:**

This is a well written paper addressing a challenging problem with an original approach.  While one reviewer claims there is not a strong call for calibration of regression tasks, this may well be because methods don't exist.  Certainly, calibration is a critical tool for classification.

The major failing of the paper, however, is the empirical evaluation.  Given that no prior work exists, it is arguably OK to not do this, but one could easily reject the paper on this issue alone, as AnonReviewer4 was inclined to do.  One reviewer, however, thought highly of the paper, which bumped up its average score, more than I think it should have got (due to the poor experimental work).

The abstract could be improved by mentioning the use of kernels, the nature of this solution is a substantial part of the paper.

---

> ### Author Response · Authors · 2021-03-17
> **Camera-ready version**
>
> We extended the abstract as suggested by mentioning the kernel calibration error.
>
> Additionally, we improved the paper in the following ways (see https://github.com/devmotion/Calibration_ICLR2021/pull/1 for a detailed changelog):
> - We fixed multiple typos (mostly in the supplementary material which is not separated from the main text anymore)
> - We updated the experiments (does not affect the results and explanations):
>    * We updated Julia package dependencies resulting in better visualizations and major performance improvements (particularly helpful for estimating p-values of the ensembles which was missing in the initial submission)
>    * We ensured reproducibility by pinning package versions and providing nix project environments
> - We added a webpage that displays automatically generated HTML output and Jupyter notebooks of the experiments (https://devmotion.github.io/Calibration_ICLR2021)
>
> Moreover, we made the calibration evaluation methods proposed in our paper (and ECE) available in the Julia packages CalibrationErrors.jl, CalibrationTests.jl, and CalibrationErrorsDistributions.jl. An (initial) documentation of the Julia packages is available at https://devmotion.github.io/CalibrationErrors.jl/dev/. Additionally, we published a Python wrapper called pycalibration that provides an interface for these packages (https://github.com/devmotion/pycalibration). Hopefully, these tools are useful for practitioners and can increase the practical relevance of our research.
>
> *Edit:* We also published an R interface at https://github.com/devmotion/rcalibration.